# Towards robust unlearnable examples via deep hiding

## Abstract

Ensuring data privacy and protection has become paramount in the era of deep learning. Unlearnable examples are proposed to mislead the deep learning models and prevent data from unauthorized exploration by adding small perturbations to data. However, such perturbations (*e.g.*, noise, texture, color change) predominantly impact low-level features, making them vulnerable to countermeasures like adversarial training, data augmentations, and preprocessing. In contrast, semantic images with intricate shapes have a wealth of high-level features, making them more resilient to countermeasures and potential for producing robust unlearnable examples. In this paper, we propose a Deep Hiding (DH) scheme that adaptively hides semantic images enriched with high-level features. We employ an Invertible Neural Network (INN) to invisibly integrate predefined images, inherently hiding them with deceptive perturbations. To enhance data unlearnability, we introduce a Latent Feature Concentration module, designed to work with the INN, regularizing the intra-class variance of these perturbations. To further boost the robustness of unlearnable examples, we design a Semantic Images Generation module that produces hidden semantic images. By utilizing similar semantic information, this module generates similar semantic images for samples within the same classes, thereby enlarging the inter-class distance and narrowing the intra-class distance. Extensive experiments on CIFAR-10, CIFAR-100, and ImageNet-subset, against 12 countermeasures, reveal that our proposed method exhibits state-of-the-art robustness for unlearnable examples, demonstrating its efficacy in data protection.

## 1 Introduction

The rapid growth of deep learning is largely attributed to the vast amounts of "free" data available on the internet. However, a significant portion of these datasets might encompass personal information obtained without clear authorization (Mahajan et al., 2018; Prabhu & Birhane, 2020). Such practices have heightened societal concerns regarding the potential misuse of individual data, particularly when leveraged to develop commercial or potentially malicious models absent the owner's consent (Drachsler & Greller, 2016). To address these concerns, the concept of unlearnable examples (Shen et al., 2019; Feng et al., 2019; Huang et al., 2021; Tao et al., 2021; Fowl et al., 2021) was introduced, which aims to prevent a deep learning model's ability to discern meaningful features from genuine patterns by introducing minor perturbations to clean images.

When we deploy unlearnable examples to protect unauthorized data in the real world, their general robustness against different countermeasures plays a critical role. Existing methods (Fu et al., 2021; Wang et al., 2021; Wen et al., 2022) mainly focus on improving their robustness against adversarial perturbation, since the unlearnable examples like error-minimization (Huang et al., 2021) or targeted adversarial poison (Fowl et al., 2021) show vulnerability to adversarial training. However, the general robustness of unlearnable examples against various countermeasures (*e.g.*, data augmentations, data preprocessing) has been ignored. For example, (Liu et al., 2023) reveals simple JPEG compression and grayscale transformation can significantly impact the effectiveness of most existing unlearnable examples methods; OPS (Wu et al., 2022) demonstrates strong adversarial robustness, while it is extremely fragile to widely used operations including cutout and median filtering.

Consequently, we introduce a Deep Hiding scheme, termed DH, devised to generate generally robust unlearnable examples for fortified data protection. Several studies (Geirhos et al., 2018; Zeiler

& Fergus, 2014; He et al., 2016a; Li et al., 2020) indicate that the natural image with semantic information (*e.g.*, intricate shapes) is robust against adversarial perturbation, data augmentations, and data preprocessing. Additionally, the existing image hiding techniques (Baluja, 2017; Yu, 2020; Jing et al., 2021a; Zhang et al., 2019; Pan et al., 2021) support adaptively hiding one image within another. Among them, the Invertible Neural Networks (INNs) (Jing et al., 2021a; Guan et al., 2022a; Xiao et al., 2023; Meng et al., 2022) are notable for their outstanding capability to render images virtually invisible.

Specifically, our proposed method employs an INN model to invisibly and adaptively hide semantic images, endowed with rich high-level attributes, into clean images, generating deceptive perturbations. To enhance the effectiveness of unlearnable examples, we introduce the Latent Feature Concentration module (LFC) to limit intra-class variance by regularizing the latent feature distance of the perturbations. Additionally, we design a Semantic Images Generation module (SIG) to produce hidden semantic images, by controlling the semantic features (*i.e.*, shapes, edges) during the generation process. Capitalizing on similar semantic information, this module generates analogous semantic images for samples within identical categories. These modules increase the inter-class separation and minimize the intra-class variance, enhancing the robustness of unlearnable examples.

In our designed scheme, the deep learning model prioritizes the features of hidden semantic images over those of genuine patterns due to the semantic nature of the hidden features. Additionally, semantic images with complex shapes possess rich high-level attributes that exhibit greater resistance to data countermeasures. In the experiments, we implemented two settings of hidden semantic images: class-wise and sample-wise, aligning them to a single class to strike a balance between efficiency and exposure risk. Extensive experiments show that our deep hiding strategy effectively conceals robust and general unlearnable examples. Across countermeasures, the ResNet-18 (He et al., 2016b) models trained on the perturbed CIFAR-10, CIFAR-100 and ImageNet-subset have average test accuracy of 16.31%, 6.47% and 8.15% respectively, compared to the best performance of 33.82%, 20.62% and 22.89% by the other unlearnable examples techniques. Our contributions can be summarized as:

- We conceptualize the generation process of unlearnable examples in data protection as an image-hiding challenge. To address this, we introduce a Deep Hiding scheme that invisibly and adaptively hides semantic images, enriched with high-level attributes, into clean images using an INN model.
- We propose the Latent Feature Concentration module, designed to regularize the intra-class variance of perturbations, enhancing the effectiveness of unlearnable examples. Moreover, we design the Semantic Images Generation module to generate hidden semantic images by maintaining semantic feature consistency within a single class, aiming to amplify the robustness of unlearnable examples.
- Extensive experiments on CIFAR-10, CIFAR-100, and ImageNet subset demonstrate that our proposed deep hiding scheme can generate notably robust unlearnable examples, which achieve state-of-the-art robust generalization on various countermeasures.

## 2 RELATED WORK

**Unlearnable examples.** To safeguard data from unauthorized scraping, there is an emerging research emphasis on techniques to render data "unlearnable" for machine learning models. Considering the surrogate models utilized in training, denoted as surrogate-dependent models, Targeted Adversarial Poisoning (TAP) (Fowl et al., 2021) employs adversarial examples as a more effective form of data poisoning, aiming to ensure that models trained on adversarially perturbed data fail to identify even their original counterparts. Building on this, Error-Minimizing (EM) (Huang et al., 2021) introduces the concept of "unlearnable examples" and employs "error-minimizing noise" through a bi-level optimization process to make data unlearnable. However, this approach is not robust against adversarial training. To address this limitation, Robust Error-Minimizing (REM) (Fu et al., 2021) introduces a robust error-minimizing noise by incorporating adversarial training and the expectation over transformation (Athalye et al., 2018) technique. Further enhancing the utility of unlearnable examples, ADVersarially Inducing Noise (ADVIN) (Wang et al., 2021) and Entangled Features (EntF) (Wen et al., 2022) propose similar methods to enhance the robustness of adversarial training. On another front, Transferable Unlearnbale Examples (TUE) (Ren et al., 2022) proposes

the classwise separability discriminant to improve their transferability across different training settings and datasets. However, the generated perturbation derived from gradient learning strongly requires knowledge from the surrogate model. In contrast, Autoregressive (AR) (Sandoval-Segura et al., 2022) introduces a surrogate-free methodology, proposing AR perturbations that remain independent of both data and models. Besides, Linear separable Synthetic Perturbations (LSP) (Yu et al., 2022) investigates the underlying mechanisms of availability attacks, identifying that the perturbations serve as "shortcuts" for learning objectives, and introducing synthetic shortcuts by sampling from Gaussian distributions. Another novel approach is One Pixel Shortcut (OPS) (Wu et al., 2022), a single pixel in each image results in significant degradation of model accuracy.

**Robustness.** Currently, certain data processing can diminish the efficacy of the added perturbation. To evaluate the robustness of these generated unlearnable examples, Image Shortcut Squeezing (ISS) (Liu et al., 2023) utilizes fundamental countermeasures based on image compression techniques, such as grayscale transformation, JPEG compression, and bit-depth reduction (BDR), to counteract the effects of perturbations. In addition, techniques such as Gaussian blur, random cropping and flipping, CutOut (DeVries & Taylor, 2017), CutMix (Yun et al., 2019), and MixUp (Zhang et al., 2017) are employed to assess the robustness of unlearnable examples. More contemporarily, AVATAR (Dolatabadi et al., 2023) extends the methodology outlined in DiffPure (Nie et al., 2022), using diffusion models to counteract intentional perturbations while preserving the essential semantics of training images. Additionally, as referenced in the unlearnable examples part, adversarial training (AT) stands as a pivotal method to bolster the resilience of crafted unlearnable examples. Notably, it's been recognized (Fu et al., 2021; Wang et al., 2021; Wen et al., 2022) as the preeminent strategy to render perturbations ineffective.

**Image hiding.** As deep learning continues to evolve, researchers are exploring methods to seamlessly embed whole images within other images using deep neural networks (Baluja, 2017; Yu, 2020; Zhang et al., 2019; Pan et al., 2021). Leveraging the inverse property of INN for image-to-image tasks (Zhao et al., 2021; Huang & Dragotti, 2022), HiNet (Jing et al., 2021b) and DeepMIH (Guan et al., 2022b) employ DWT to decompose the input image, and constrain the hiding to implementation in high-frequency sub-bands for invisible hiding. Similarly, iSCMIS (Li et al., 2023), ISN (Lu et al., 2021), and RIIS (Xu et al., 2022) hide data by using the inverse property, with some models even simulating data transformations to enhance the robust retrieval of hidden data. In the backdoor and adversarial attack fields, image hiding schemes have notably contributed. Specifically, Backdoor Injection attack (Zhong et al., 2020) and Poison Ink (Zhang et al., 2022) subtly embed perturbation masks and image structures into training data as their trigger patterns, respectively, to mislead the model into misclassifying instances with the backdoor to a target label. AdvINN (Chen et al., 2023) utilizes INN to generate inconspicuous yet resilient adversarial examples, offering superior stealth and robustness over conventional methods. Such strategies underscore the significant potential of deep image hiding in bolstering the effectiveness of unlearnable examples.

## 3 PROPOSED METHOD

### 3.1 DEFINITION

**Recalling unlearnable examples.** Following the existing unlearnable research (Huang et al., 2021; Fu et al., 2021; Fowl et al., 2021; Sandoval-Segura et al., 2022; Yu et al., 2022), we focus on the image classification task in this work. Given a clean dataset $\mathcal{D}_c = \{(\boldsymbol{x}_i, y_i)\}_{i=1}^n$ with $n$ training samples, where $\boldsymbol{x} \in \mathcal{X} \subset \mathbb{R}^d$ represents the images and $y \in \mathcal{Y} = \{1, \cdots, K\}$ denotes its corresponding labels. We assume an unauthorized party will use a classifier given as $f_\theta : \mathcal{X} \to \mathcal{Y}$ where $\theta \in \Theta$ is the classifier parameters. To safeguard the images from unauthorized training, rather than publishing $\mathcal{D}_c$, existing methods (Huang et al., 2021; Fu et al., 2021) introduce perturbation to clean images, generating an unlearnable dataset as:

$$\mathcal{D}_u = \{(\boldsymbol{x}_i + \boldsymbol{\delta}_i, y_i)\}_{i=1}^n, \tag{1}$$

where $\boldsymbol{\delta}_i \in \Delta_{\mathcal{D}} \subset \mathbb{R}^d$ and $\Delta_{\mathcal{D}}$ is the perturbation set for $\mathcal{D}_c$. The objective of unlearnability is to ensure that a classifier $f_\theta$ trained on $\mathcal{D}_u$ exhibits poor performance on test datasets.

**Proposed unlearnable examples.** Current approaches typically generate perturbations either through gradient-based training with a surrogate model or by sampling noise from a predefined dis-

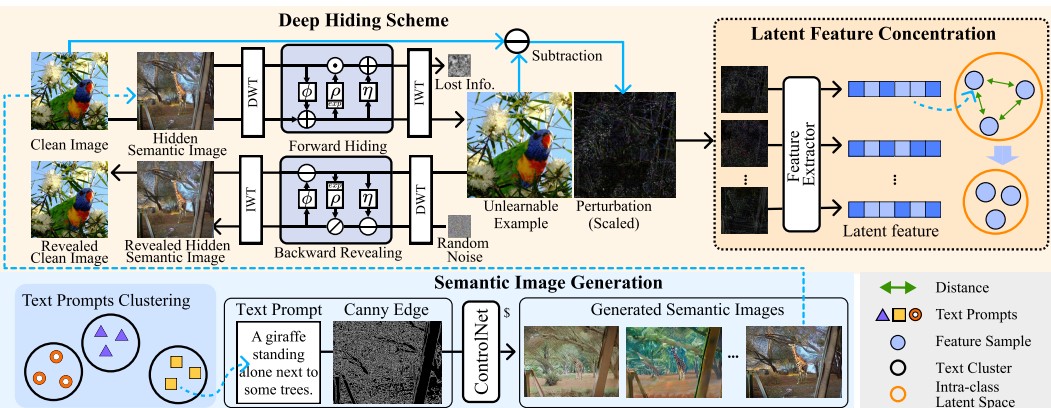

Figure 1: Overall pipeline of the proposed method. A generative model is employed to generate the hidden semantic images. These generated images are then hidden within clean images using a Deep Hiding model. The Latent Feature Concentration module is designed to constrain the intra-class variance by regularizing the latent feature distance of perturbations.

tribution in model-free manners. These perturbations lack semantic high-level features and redundancy, making it challenging to generalize robustness against various countermeasures. Conversely, we propose an adaptive method for embedding a semantic image $\boldsymbol{h}_i$ characterized by rich high-level features, within a clean image to generate unlearnable examples. Thus the generated unlearnable dataset is defined as:

$$\mathcal{D}_u = \left\{ (\mathcal{F}(\boldsymbol{x}_i, \boldsymbol{h}_i), y_i) \right\}_{i=1}^n, \qquad (2)$$

where $\mathcal{F}(\cdot, \cdot)$ represents our hiding model. We adaptively hide predefined semantic images into clean datasets $\mathcal{D}_c$ rather than arbitrary perturbation, inherently introducing deceptive perturbations to mislead classifier $f_\theta$, thereby enhancing the effectiveness of unlearnable examples.

## 3.2 DEEP HIDING SCHEME FOR ROBUST UNLEARNABLE EXAMPLES

To generate a resilient unlearnable dataset $\mathcal{D}_u$, we introduce the Deep Hiding scheme. This framework incorporates the image hiding model, which integrates the specially-designed Latent Feature Concentration module, and the Semantic Images Generation module. Figure 1 illustrates the overview of the proposed Deep Hiding scheme.

### 3.2.1 DEEP HIDING MODEL

Inspired by the image hiding methods (Baluja, 2017; Yu, 2020; Jing et al., 2021a; Zhang et al., 2019; Pan et al., 2021), we employ the INN-based hiding model, HiNet (Jing et al., 2021b), as our framework, leveraging its superior generation performance. HiNet employs $N$ affine coupling blocks to form two invertible processes, forward hiding and backward revealing, where the hiding process enables inherently embedding predefined semantic images into clean images, as illustrated in Figure 1. To facilitate invisible deep hiding, Discrete Wavelet Transform (DWT) $\mathcal{T}(\cdot)$ is applied to decompose the input clean image $\boldsymbol{x}_c$, and hidden semantic image $\boldsymbol{x}_h$ into low and high-frequency sub-bands. We denote the sub-bands features of $\boldsymbol{x}_c$ and $\boldsymbol{x}_h$ as $\boldsymbol{z}_c = \mathcal{T}(x_c)$ and $\boldsymbol{z}_h = \mathcal{T}(x_h)$, respectively. Let $\boldsymbol{z}_c^i$ and $\boldsymbol{z}_h^i$ be the input features of the $i^{th}$ affine coupling block, the forward hiding process of this block can be expressed as:

$$\boldsymbol{z}_c^i = \boldsymbol{z}_c^{i-1} + \phi\left(\boldsymbol{z}_h^{i-1}\right), \text{and } \boldsymbol{z}_h^i = \boldsymbol{z}_h^{i-1} \odot \exp\left(\alpha \cdot \rho\left(\boldsymbol{z}_c^i\right)\right) + \eta\left(\boldsymbol{z}_c^i\right), \qquad (3)$$

where $\phi(\cdot)$, $\rho(\cdot)$, and $\eta(\cdot)$ are three sub-modules, sharing the same network structure but with different weights, $\exp(\cdot)$ is the Exponential function, $\odot$ is the Hadamard product operation, and $\alpha$ controls the weight of exponential operation. Given the output features $\boldsymbol{z}_c^N$ of total $N^{th}$ affine coupling block, the unlearnable examples $\boldsymbol{x}_{ue} = \mathcal{T}^{-1}(\boldsymbol{z}_c^N)$ are generated by Inverse DWT (IDWT).

To ensure the success of the image-hiding procedure, in the backward revealing process, the obtained unlearnable examples are first decomposed by DWT and then together with the randomly sampled latent noises $r$ feed into the HiNet, resulting in the revealed clean image $\boldsymbol{x}_c' = \mathcal{T}^{-1}(\boldsymbol{z}_c^1)$ and revealed

hidden semantic image $x'_h = \mathcal{T}^{-1}(r^1)$ by subsequent IDWT. Such $z^1_c$ and $r^1$ can be obtained by:

$$z^{i-1}_c = \left(z^i_c - \eta\left(z^i_c\right)\right) \odot \exp\left(-\alpha \cdot \rho\left(z^i_c\right)\right), \text{ and } r^{i-1} = r^i - \phi\left(r^{i-1}\right), \tag{4}$$

Our primary objective is to generate invisible unlearnable examples. To ensure this, we restrict them to a specific radius $\epsilon$ of perturbation, characterized by the hiding loss as:

$$\mathcal{L}_{\text{hide}}\left(x_{ue}, x_c\right) = \max\left(\text{MSE}(x_{ue}, x_c), \epsilon^2\right), \tag{5}$$

For consistency and fairness, we adopt the same radius $\epsilon = 8/255$ as utilized in existing unlearnable examples methodologies (Huang et al., 2021; Fu et al., 2021; Fowl et al., 2021).

In addition, we adapt the loss functions from HiNet (Jing et al., 2021a) to concurrently ensure optimal image hiding performance. Consequently, the total loss for the Deep Hiding module is represented as follows:

$$\mathcal{L}_{\text{DH}} = \mathcal{L}_{\text{hide}}\left(x_{ue}, x_c\right) + \omega_1 \cdot \mathcal{L}_{\text{freq}}\left(\mathcal{H}\left(x_{ue}\right)_{LL}, \mathcal{H}\left(x_c\right)_{LL}\right) + \omega_2 \cdot \mathcal{L}_{\text{reveal}}\left(x'_h, x_h\right), \tag{6}$$

As verified by (Jing et al., 2021a; Guan et al., 2022a), information hidden in high-frequency components is less detectable than in low-frequency ones. To optimize the anti-detection and invisibility of unlearnable examples, it's crucial to maintain the low-frequency sub-bands to closely resemble those of clean images. $\mathcal{L}_{\text{freq}}$ measures the $\ell_2$ distance between the low-frequency sub-bands of clean images and unlearnable examples, further bolstering the stealthiness. $\mathcal{H}(\cdot)_{LL}$ is the function of extracting low-frequency sub-bands after wavelet decomposition. Additionally, $\mathcal{L}_{\text{reveal}}\left(x'_h, x_h\right)$ measures the $\ell_2$ distance between revealed hidden images $x'_h$ and hidden semantic images $x_h$, ensuring the success of the image hiding process.

### 3.2.2 LATENT FEATURE CONCENTRATION

Although the deep hiding model effectively embeds high-level features from predefined semantic images into clean images, delivering outstanding unlearnability (see Section 4.2), the adaptive hiding process still results in latent features of perturbations with non-uniform intra-class distribution. A compact distribution of these latent features could significantly mislead the learning trajectory of DNNs, by offering a distinct learning shortcut across similar intra-class images. To address this, we introduce the Latent Feature Concentration module, specifically designed to regularize the intra-class variance of perturbations, further boosting the effectiveness of unlearnable examples. The perturbation represents the variation between the generated unlearnable example and its corresponding clean image, defined as $x_{pm} = x_{ue} - x_c$. We utilize a pre-trained ResNet-18 (He et al., 2016b) as the feature extractor, denoted by $\mathcal{G}(\cdot)$. The latent features are extracted from the output final convolution layer. Our objective is to minimize the variation between latent features derived from the perturbation maps for images within the same class. Consequently, the concentration loss $\mathcal{L}_{\text{conc}}$ is represented as:

$$\mathcal{L}_{\text{conc}} = \sum_{i,j,y^i=y^j} dis(\mathcal{G}(x^i_{pm}), \mathcal{G}(x^j_{pm})), \tag{7}$$

where $dis(\cdot, \cdot)$ denotes the cosine distance between the two flattened latent features, and $y$ represents the label. Thus, the total loss of our proposed method is described as:

$$\mathcal{L}_{\text{total}} = \mathcal{L}_{\text{DH}} + \omega_3 \cdot \mathcal{L}_{\text{conc}}. \tag{8}$$

### 3.2.3 SEMANTIC IMAGES GENERATION

Though the deep hiding model can embed human-imperceptible perturbations, it can not ensure efficacy when the hidden images are randomly picked. Consequently, we introduce a generative method specifically designed for controlled hidden semantic image generation, aiming to achieve desired intra-class and inter-class distributions; that is, a distinct inter-class distance complemented by a minimal intra-class distance. As shown in Figure 1, we use pre-trained generative models, *i.e.*, Stable Diffusion (Rombach et al., 2022) and ControlNet (Zhang & Agrawala, 2023), to generate hidden semantic images by controlling both text prompts and canny edge maps. These text prompts, sourced from (Gandikota et al., 2023), characterize images from the COCO datasets (Lin et al., 2014). The canny edge maps are derived by applying the canny filter to the corresponding images.

To maximize the inter-class distance among hidden semantic images, we choose text prompts that exhibit the greatest variation from the rest. We first cluster all text prompts using K-Means (Arthur

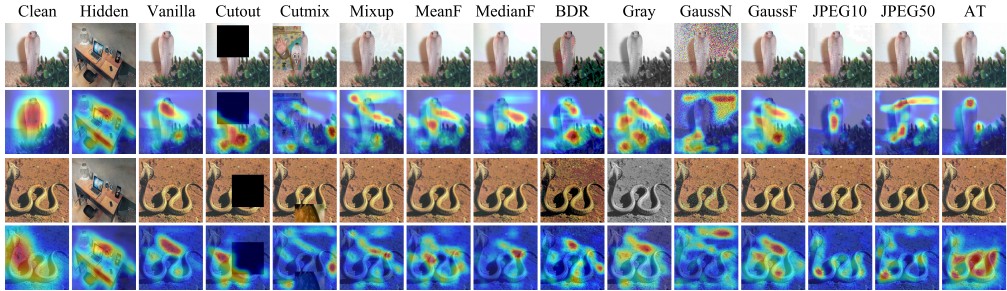

Figure 2: Grad-CAM visualization of unlearnable examples derived from the ImageNet subset under different countermeasures. Note that Red regions typically indicate the areas the model paid the most attention to, while Blue regions colors indicate less attention.

& Vassilvitskii, 2007) based on their semantic features via CLIP text extractor (Radford et al., 2021). Subsequently, we identify the top-$k$ distinct clusters, where $k$ represents the number of classes in the targeted dataset $\mathcal{D}_c$. In each of these clusters, we choose the text prompt nearest to the cluster center, which represents a unique semantic feature. To minimize the intra-class distance among hidden semantic images, we ensure their consistency in high-level features by controlling the key image attributes, *i.e.*, shapes. Consequently, we obtain a canny edge map of the text-corresponding image, which acts as the condition for ControlNet. Then, we use the Stable Diffusion model and ControlNet (SD+C) to generate semantic images as the hidden semantic input $x_h$ for our DH scheme. With these specifically generated hidden semantic images, our deep hiding model can guarantee the general robustness of the unlearnable examples.

### 3.3 Properties of Deep Hiding Scheme

DNNs are capable of learning complex features for image understanding. However, they are inclined to overfit to the patterns that are much *easier* to learn (Geirhos et al., 2020), in alignment with the "Principle of Least Effort" (Zipf, 2016). With this phenomenon, many unlearnable examples are proposed to protect the data from being learned by DNNs. Consequently, DNNs predominantly focus on misleading perturbations rather than the intended solutions. Our Deep Hiding scheme exploits the same

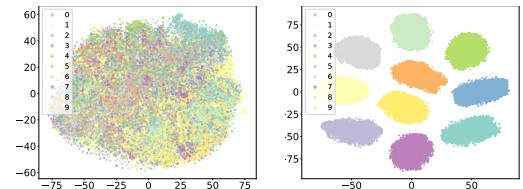

Figure 3: The t-SNE visualization of the models' feature representations on the clean samples (left) and the perturbation generated by our DH scheme (right) on CIFAR-10.

principle. In our proposed scheme, clean images within a given class are embedded with similar hidden semantic images that share the same global shape but differ in their local textures. Compared to the complex features in the original images, the embedded similar semantic information is much *easier* to be learned by DNNs. The visual representation in Figure 3 demonstrates that the perturbations generated by our scheme exhibit clear separability, marked by straightforward decision boundaries. Besides, We utilize Grad-CAM (Selvaraju et al., 2017) to visualize the attention of DNNs in Figure 2. It is obvious that the attention is redirected toward the desk (the hidden semantic image) rather than the snake (the clean image) during classification. While DNNs can take non-semantic features as "shortcuts" for more effortless learning, these features are vulnerable to simple data augmentations and data processing. Different from the existing unlearnable examples methods, we incorporate natural images as hidden semantic images to generate perturbations. These perturbations, enriched with deep high-level semantic attributes, exhibit robustness against diverse countermeasures. As illustrated in Figure 2, the hidden semantic information can mislead the DNNs to similar key regions after most countermeasures. These findings affirm the efficacy and resilience of using natural images as hidden semantic information when faced with various countermeasures.

## 4 Experiments

### 4.1 Experimental Setups

**Datasets and models.** We use three image classification datasets, CIFAR-10 (Krizhevsky et al., 2009), CIFAR-100 (Krizhevsky et al., 2009), and 100-class subset of ImageNet (Deng et al., 2009)

in our experiments. We evaluate on the ResNet-18 (He et al., 2016b) in our main experiments. To study the transferability of the proposed DH scheme, we consider models with diverse architectures, including ResNet-50 (He et al., 2016b), VGG-19 (Simonyan & Zisserman, 2014), DenseNet-121 (Huang et al., 2017), and ViT (Dosovitskiy et al., 2020).

**DH model training.** Our training exclusively utilizes the ImageNet subset comprising 100 classes for the DH model. As detailed in Section 3.2.3, for each class, we generate 100 semantic images using paired text prompts and canny edge maps. Our training configuration is as follows: 5k iterations, the Adam optimizer (Kingma & Ba, 2014) with hyper-parameters set at $\beta_1 = 0.5$, $\beta_2 = 0.999$, and $\epsilon = 10^{-6}$; a consistent learning rate of $1 \times 10^{-4.5}$; and a mini-batch size of 24. To ensure stable model training, we assign weights of 1 to $\omega_1$ and $\omega_2$, and a weight of 0.0001 to $\omega_3$ across all experiments. Subsequent to this, the pre-trained DH model is used to generate unlearnable examples across the three datasets: CIFAR-10, CIFAR-100, and the ImageNet subset. The unlearnable examples generation follows two settings: *class-wise setting and sample-wise setting*. In the class-wise setting, we hide a consistent semantic image in the clean images of each class; whereas in the sample-wise setting, the hidden semantic images in each class are sampled from the generative model with the same text prompt and canny edge map, so they share the same global shape but differ in their local textures.

**Classifier training.** To evaluate the effectiveness of the generated unlearnable examples, we employ a standard classification problem. For both CIFAR-10 and CIFAR-100 datasets, we follow the official train-test division. Regarding the ImageNet subset, we allocate 20% of images from the first 100 classes of the official ImageNet training set for training purposes, using all related images in the official validation set for testing.

**Compared methods.** We compare the proposed deep hiding scheme with six state-of-the-art unlearnable examples methods, including four surrogate-dependent methods, EM (Huang et al., 2021), REM (Fu et al., 2021), TAP (Fowl et al., 2021), and EntF (Wen et al., 2022), and three surrogate-free methods, AR (Sandoval-Segura et al., 2022), LSP (Yu et al., 2022), and OPS (Wu et al., 2022). Note that we re-implement all the methods based on the public available codes.

**Robustness evaluation.** To evaluate the robustness of our generated unlearnable examples, we train ResNet-18 with them across a variety of data augmentations and preprocessing methods, as suggested in (Liu et al., 2023). For augmentation, we employ vanilla (random crop and resize), cutout (DeVries & Taylor, 2017), cutmix (Yun et al., 2019), and mixup (Zhang et al., 2017). Additionally, we utilize seven data preprocessing techniques: Mean Filter (MeanF), Median Filter(MedianF), Bit-Depth Reduction (BDR), Grayscale transformation (Gray), Gaussian Noise (GaussN), Gaussian Filter (GaussF), and JPEG compression. Additionally, we also implement Adversarial Training (AT). Notable parameter settings include: JPEG compression qualities set at 10% and 50%, an AT radius of $8/255$, and a Gaussian noise distribution of $\mathcal{N}(0, 0.1)$. More detailed evaluation procedures can be referenced in the supplemental material.

## 4.2 EFFECTIVENESS OF THE PROPOSED METHOD.

We first evaluate the efficacy of the proposed method by training ResNet18 with unlearnable examples and testing on clean images. In Table 1, we present detailed test accuracy results across three datasets: CIFAR-10, CIFAR-100, and the ImageNet subset. Notably, both our class-wise and sample-wise settings consistently achieve state-of-the-art performance on all three datasets. Specifically, the results of the vanilla training show that our class-wise setting degrades the test accuracy to 10%, 1.47%, and 1.02% for three datasets respectively, which are nearly random-guessing. It indicates that the models can not learn any useful semantic information for the classification task once we hide the semantic images into clean images. In contrast, the other unlearnable examples techniques fail to maintain consistent performance across datasets, For instance, both EM and LSP result in much higher test accuracies on ImageNet. Even though we use a sample-wise setting to reduce the exposure risk of the hidden image, it still achieves promising performance across datasets, especially on ImageNet. We hypothesize that our unlearnable examples carry abundant information due to their semantic image nature, making them more generally effective in various scenarios. Besides, we further visualize the perturbations generated by our hiding scheme, as shown in Figure 4. Compared to the perturbations provided by other methods, our perturbation consistently exhibits high-level features (*e.g.*, shape) that align with the hidden semantic image. These findings underscore the effectiveness of our proposed Deep Hiding approach.

Table 1: Test accuracy (%) of models trained on unlearnable examples from CIFAR-10, CIFAR-100, and ImageNet subset against data augmentations, data preprocessing, and adversarial training. Numbers in **Bold** and underline numbers indicate the best and second-best results, respectively.

| | Method | Vanilla | Cutout | Cutmix | Mixup | MeanF | MedianF | BDR | Gray | GaussN | GaussF | JPEG10 | JPEG50 | AT | Mean |
|---|---|---|---|---|---|---|---|---|---|---|---|---|---|---|---|
| CIFAR-10 | Clean | 94.59 | 95.00 | 94.77 | 94.96 | 49.70 | 86.64 | 89.07 | 92.80 | 88.71 | 94.54 | 85.22 | 90.89 | 84.19 | 87.78 |
| | EM | 10.10 | **10.00** | 15.39 | 16.82 | 10.63 | 24.27 | 35.90 | 69.29 | 32.96 | 10.01 | 84.80 | 87.82 | 84.28 | 37.87 |
| | REM | 29.00 | 29.42 | 26.13 | 28.37 | 19.07 | 32.80 | 39.93 | 69.83 | 39.97 | 28.67 | 84.15 | 77.65 | 85.93 | 45.46 |
| | TAP | 25.90 | 32.69 | 26.77 | 40.46 | 31.68 | 65.12 | 80.25 | 26.36 | 88.66 | 26.09 | 84.77 | 90.31 | 83.57 | 56.39 |
| | EntF | 91.50 | 91.30 | 90.93 | 92.52 | 17.85 | 70.28 | 91.46 | 80.33 | 90.31 | 79.79 | 74.36 | 83.56 | 75.86 | 79.23 |
| | LSP | 19.07 | 19.87 | 20.89 | 26.99 | 28.85 | 29.85 | 66.19 | 82.47 | 19.25 | 16.19 | 83.01 | 57.87 | 84.59 | 42.70 |
| | AR | 13.31 | 11.35 | 12.21 | 13.30 | 12.38 | 17.04 | 37.42 | 34.81 | 42.29 | 12.56 | 85.08 | 89.63 | 58.23 | 33.82 |
| | OPS | 16.53 | 89.73 | 83.91 | 34.88 | 17.31 | 86.86 | 43.04 | 16.65 | 36.72 | 15.10 | 82.79 | 57.00 | **9.42** | 45.38 |
| | Ours(S) | 15.36 | 10.79 | **10.00** | 14.72 | 17.68 | 17.00 | 21.12 | 17.61 | 22.78 | 11.16 | 80.41 | 81.03 | 38.31 | 27.54 |
| | Ours(C) | **10.00** | **10.00** | 11.25 | **10.02** | **10.59** | **10.04** | **13.53** | **10.00** | **10.00** | **10.00** | **72.97** | **23.62** | 10.00 | **16.31** |
| CIFAR-100 | Clean | 75.82 | 74.45 | 76.32 | 77.07 | 14.72 | 50.72 | 63.51 | 70.04 | 62.41 | 75.86 | 57.35 | 68.59 | 58.25 | 62.44 |
| | EM | 2.84 | 12.05 | 7.67 | 12.86 | 13.52 | 43.61 | 62.12 | 62.37 | 62.01 | 73.47 | 57.29 | 67.50 | 57.89 | 41.17 |
| | REM | 7.13 | 10.32 | 11.25 | 8.65 | 5.90 | 12.31 | 19.95 | 48.48 | 26.27 | 7.32 | 57.15 | 65.10 | 58.9 | 26.06 |
| | TAP | 14.00 | 16.55 | 15.99 | 22.56 | 5.86 | 31.95 | 55.12 | 8.90 | 61.4 | 13.95 | 56.56 | 66.67 | 56.53 | 32.77 |
| | EntF | 72.55 | 69.65 | 70.68 | 73.81 | 8.67 | 36.87 | 55.22 | 67.00 | 58.54 | 73.10 | 51.42 | 63.69 | 52.44 | 57.97 |
| | LSP | 2.68 | 2.55 | 2.69 | 4.39 | 7.15 | 6.76 | 28.23 | 42.77 | 22.42 | 2.19 | 55.23 | 33.60 | 57.45 | 20.62 |
| | AR | 1.50 | 1.47 | 1.56 | **1.37** | 5.35 | 3.89 | 28.28 | 19.68 | 59.34 | 1.57 | 56.99 | 65.72 | 58.33 | 23.47 |
| | OPS | 11.69 | 71.36 | 64.25 | 12.59 | 3.18 | 49.74 | 19.31 | 18.70 | 17.30 | 11.79 | 56.72 | 48.71 | 10.22 | 30.43 |
| | Ours(S) | 4.79 | 4.13 | 5.39 | 4.72 | 6.22 | 10.21 | 12.12 | 3.72 | 19.85 | 3.61 | 49.50 | 34.86 | 41.12 | 15.40 |
| | Ours(C) | **1.47** | **1.03** | **1.06** | 1.47 | **1.04** | **1.45** | **1.72** | **1.38** | **1.08** | **1.00** | **44.58** | **25.45** | **1.39** | **6.47** |
| ImageNet subset | Clean | 63.93 | 64.02 | 55.10 | 64.55 | 19.92 | 36.08 | 56.63 | 68.35 | 50.62 | 65.40 | 56.83 | 69.36 | 48.24 | 55.31 |
| | EM | 28.99 | 18.78 | 17.61 | 36.55 | 7.46 | 32.60 | 53.43 | 17.93 | 44.63 | 26.04 | 53.41 | 56.96 | 43.56 | 33.69 |
| | REM | 14.78 | 14.10 | 11.73 | 19.88 | 15.32 | 14.12 | 16.48 | 44.74 | 15.96 | 15.34 | 50.50 | 17.14 | 47.52 | 22.89 |
| | TAP | 7.96 | 15.02 | 15.18 | 23.08 | 10.44 | 15.02 | 47.97 | 22.93 | 46.84 | 12.80 | 53.40 | 37.98 | 44.18 | 27.14 |
| | LSP | 18.18 | 9.52 | 34.16 | 9.76 | 4.14 | 5.20 | 43.38 | 52.66 | 34.28 | 17.92 | 51.80 | 49.06 | **42.26** | 28.64 |
| | Ours(S) | 3.36 | 2.14 | 3.30 | 2.32 | 2.52 | 6.48 | 8.62 | 1.32 | 4.02 | 1.94 | 39.92 | 26.66 | 44.56 | 11.32 |
| | Ours(C) | **1.02** | **1.56** | **1.28** | **1.44** | **1.14** | **1.74** | **2.94** | **1.32** | **1.70** | **0.98** | **34.78** | **12.00** | 44.00 | **8.15** |

## 4.3 ROBUSTNESS OF THE PROPOSED METHOD.

To evaluate the robustness of our generated unlearnable examples, we adopt a variety of countermeasures, including four data augmentation, seven data preprocessing techniques, and Adversarial Training (AT). As shown in Table 1, the experimental results demonstrate that the proposed method consistently outperforms the other techniques, exhibiting robust performance against all countermeasures. On the CIFAR-10 dataset, our method reduces the test accuracy to 10%~13.53% on most types of countermeasures, except JPEG compression. For the CIFAR-100 dataset, the test accuracy fluctuates

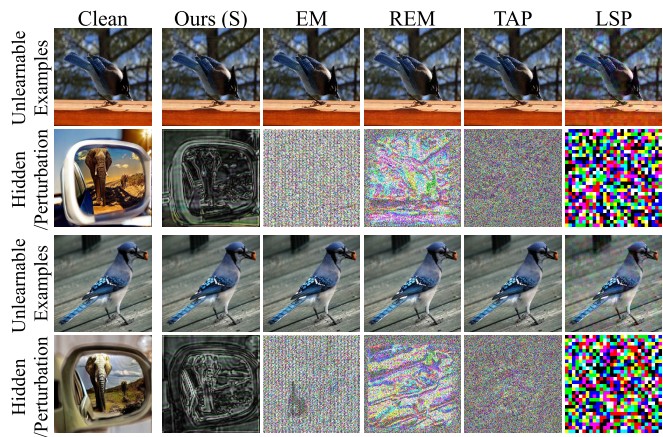

Figure 4: Visualization of perturbation maps. Perturbations are absoluted and normalized to [0,1] for a better view.

between 1% and 1.72%. It is worth noting that JPEG compression and adversarial training can suppress most types of unlearnable example techniques, also as mentioned by (Liu et al., 2023; Fu et al., 2021). While surrogate-dependent techniques, such as EM, REM, TAP, and EntF, generate high-frequency and gradient-related perturbations, making them vulnerable to information suppression methods. Especially for EntF, though it is relatively robust against adversarial training, it is not as effective as other methods in other training settings. In contrast, we find that the surrogate-free techniques including LSP, OPS and ours have relatively stronger robustness against these 2 types of countermeasures. However, OPS inherently has trouble with the cropping and filtering operation due to its one-pixel modification, leading to diminished results in scenarios like cutout, cutmix, and median filtering. Meanwhile, LSP introduces color-based patterns, thus it is fragile to grayscale manipulation. Overall, our proposed method achieves consistent performance on most countermeasures across all datasets. Even adversarial training cannot erase our perturbation, where we hypothesize that our semantic-based perturbation differs from gradient-based perturbation. Specifically, we obtain mean test accuracy of 16.31%, 6.47%, and 8.15% on CIFAR-10, CIFAR-100, and ImageNet subset, respectively, compared to the best performances of other methods at 33.82%, 23.47%, and 22.89%. The results show that our deep hiding scheme obtains a better robust generalization of unlearnable examples with high-level semantic features.

Table 2: Test accuracy (%) of CIFAR-10 and CIFAR-100 on five architectures, including ResNet-18 (R18), ResNet-50 (R50), VGG-19 (V19), and DenseNet-121 (D121), and Vision Transformer (ViT).

| Dataset | CIFAR-10 | | | | | CIFAR-100 | | | | |
|---|---|---|---|---|---|---|---|---|---|---|
| Model | R18 | R50 | V19 | D121 | ViT | R18 | R50 | V19 | D121 | ViT |
| EM | 10.10 | 10.00 | 10.82 | 12.56 | 11.88 | 2.84 | 3.88 | 9.23 | 64.87 | 7.65 |
| REM | 30.40 | 25.10 | 24.54 | 30.28 | 32.36 | 7.13 | 7.45 | 5.26 | 12.47 | 6.91 |
| TAP | 25.93 | 25.48 | 30.36 | 78.59 | 70.96 | 14.00 | 14.25 | 33.18 | 52.64 | 14.49 |
| EntF | 91.50 | 91.83 | 88.17 | 83.30 | 69.23 | 72.55 | 73.19 | 65.68 | 60.85 | 49.43 |
| LSP | 16.99 | 14.55 | 11.53 | 24.83 | 23.78 | 2.68 | 4.06 | 2.84 | 27.05 | 9.40 |
| OPS | 17.46 | 16.73 | 19.12 | 18.40 | 28.09 | 11.69 | 10.90 | 5.67 | 10.38 | 17.23 |
| AR | 11.88 | 15.83 | 13.21 | 22.28 | 19.84 | 1.50 | 2.13 | 3.48 | 19.55 | 5.69 |
| Ours(S) | 15.36 | 12.66 | 13.16 | 16.81 | 17.83 | 4.79 | 4.90 | 6.00 | 7.02 | 10.10 |
| Ours(C) | 10.00 | 10.00 | 10.00 | 10.20 | 10.01 | 1.47 | 1.00 | 0.95 | 1.00 | 1.38 |

Table 3: Test accuracy (%) of CIFAR-10 on the models trained by the clean data mixed with different percentages of unlearnable examples.

| Method | 20% | 40% | 60% | 80% |
|---|---|---|---|---|
| EM | 94.30 | 93.09 | 91.42 | 87.29 |
| REM | 93.83 | 92.69 | 91.12 | 86.92 |
| TAP | 93.82 | 92.78 | 91.96 | 88.49 |
| EntF | 93.40 | 91.71 | 91.25 | 91.07 |
| LSP | 93.50 | 92.47 | 90.21 | 84.81 |
| OPS | 93.64 | 92.63 | 90.05 | 84.42 |
| AR | 94.07 | 92.66 | 90.34 | 85.18 |
| Sample-wise | 93.53 | 92.67 | 89.99 | 84.47 |
| Class-wise | 93.73 | 92.41 | 90.08 | 84.40 |

Table 4: Ablation studies on CIFAR10 for designed Latent Feature Concentration module (LFC), and Semantic Images Generation module (SIG), including Text Prompts Clustering (TPC) and Stable Diffusion model and ControlNet (SD+C).

| Setting | LFC | SIG | | Vanilla | Cutout | Cutmix | Mixup | MeanF | MedianF | BDR | Gray | GaussN | GaussF | JPEG10 | JPEG50 | AT | Mean |
|---|---|---|---|---|---|---|---|---|---|---|---|---|---|---|---|---|---|
| | | TPC | SD+C | | | | | | | | | | | | | | |
| Ours(S) | × | × | × | 94.08 | 94.52 | 94.07 | 88.23 | 65.42 | 87.04 | 89.46 | 91.45 | 88.36 | 94.07 | 83.29 | 89.97 | 86.26 | 88.17 |
| | × | ✓ | ✓ | 14.77 | 22.20 | 13.06 | 23.44 | 30.18 | 51.43 | 35.66 | 17.31 | 37.50 | 15.80 | 81.97 | 81.48 | 81.24 | 38.93 |
| | ✓ | × | ✓ | 10.00 | 16.53 | 20.81 | 17.14 | 18.51 | 21.73 | 24.98 | 13.85 | 22.07 | 10.59 | 80.09 | 82.90 | 46.54 | 29.67 |
| | ✓ | × | × | 94.09 | 94.34 | 93.84 | 94.00 | 64.55 | 85.94 | 88.86 | 91.00 | 88.70 | 94.20 | 82.84 | 90.31 | 85.31 | 88.38 |
| | ✓ | ✓ | ✓ | 15.36 | 10.79 | 10.00 | 14.72 | 17.68 | 17.00 | 21.12 | 17.61 | 22.78 | 11.16 | 80.41 | 81.03 | 38.31 | 27.54 |
| Ours(C) | × | × | × | 12.16 | 11.02 | 11.43 | 10.81 | 10.14 | 13.28 | 10.10 | 12.12 | 10.00 | 10.00 | 78.21 | 15.32 | 12.87 | 16.73 |
| | × | ✓ | ✓ | 10.00 | 10.00 | 10.06 | 10.82 | 13.26 | 25.76 | 15.83 | 14.20 | 10.03 | 10.00 | 76.26 | 20.28 | 14.78 | 18.56 |
| | ✓ | × | ✓ | 10.00 | 10.00 | 10.00 | 12.32 | 9.72 | 10.00 | 10.00 | 10.59 | 10.00 | 10.00 | 71.44 | 30.27 | 10.00 | 16.49 |
| | ✓ | × | × | 10.00 | 10.00 | 11.41 | 10.00 | 10.01 | 11.71 | 10.00 | 10.01 | 10.00 | 10.00 | 58.38 | 10.05 | 10.00 | 13.97 |
| | ✓ | ✓ | ✓ | 10.00 | 10.00 | 11.25 | 10.02 | 10.59 | 10.04 | 13.53 | 10.00 | 10.00 | 10.00 | 72.97 | 23.62 | 10.00 | 16.31 |

## 4.4 PERFORMANCE ON DIFFERENT ARCHITECTURES AND UNLEARNABLE PERCENTAGES.

**Different model architectures.** In real-world scenarios, the protector may not know the details of the target model. In such cases, it's critical for unlearnable examples to be transferable. Hence, we evaluate the effectiveness of the proposed method across various deep learning architectures on CIFAR-10 and CIFAR-100 datasets. As shown in Table 2, our approach consistently performs well across all five models, especially in the class-wise setting.

**Different unlearnable percentages.** Consider a situation where it's not feasible to protect all the data. This scenario is realistic since a practitioner who gains access to unlearnable examples might also obtain additional clean data from other avenues. Consequently, it's common practice to evaluate unlearnable examples' efficacy by training deep learning models with a random subset of unlearnable examples. To this end, we evaluate the performance of our proposed approach by using varying mixtures of clean images and unlearnable examples, the results are shown in Table 3.

## 4.5 ABLATION STUDY

To understand the pivotal roles of LFC and SIG in our approach, we conduct an ablation study focused on unlearnable performance. The results are shown in Table 4. In the class-wise setting, we find that the improvement of the generation module is marginal; However, in the sample-wise setting, the SIG can degrade the mean accuracy from 88% to 38%. It is further reduced to 27.5% with LFC. When we disentangle the TPC and SD+C, we find that SD+C contributes the most and TPC contributes around 2% reduction. The ablation study shows that each component in our proposed method plays an important role in the generally robust unlearnable examples.

## 5 CONCLUSION

In this paper, we present a novel Deep Hiding scheme tailored for the generation of universally robust unlearnable examples. By embedding clean images with semantically rich high-level attributes, we ensure that the generated unlearnable examples effectively derail the learning processes of unauthorized deep learning models. Additionally, our uniquely conceived Latent Feature Concentration (LFC) module further enhances the effectiveness of unlearnable examples by regularizing the intra-class variance of perturbations. To guarantee the robustness of unlearnable examples, we introduce the Semantic Images Generation (SIG) module to generate hidden semantic images by maintaining semantic feature consistency within each class. The extensive experimental results demonstrate that our proposed method achieves outstanding unlearnability performance and superior robustness against common countermeasures.

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
