# A  Appendix

This section provides some the experiment details omitted from Section 4, additional experimental results, and a byproduct. We will release the code and pretrained weights once this paper is accepted.

## A.1  Experiment Details

### A.1.1  Data augmentation

The specific details are in alignment with the REM settings. For CIFAR-10 and CIFAR-100 datasets, our data augmentation comprises random flipping, padding by 4 pixels on each side, followed by random cropping to a size of $32 \times 32$. Each image's pixels are then rescaled to the range $[-0.5, 0.5]$. In the case of the ImageNet subset, we augment the data using random cropping, resizing the images to a $224 \times 224$ dimension, implementing random flipping, and then rescaling every pixel to the interval $[-0.5, 0.5]$. Additionally, we perform 40,000 iterations on the CIFAR-10 and CIFAR-100 datasets, and 8,000 iterations on the ImageNet subset.

### A.1.2  Hyperparameters for Different Countermeasures

Unless explicitly mentioned, we maintain the following settings in our experiments on different countermeasures,

**Bit-Depth Reduction (BDR).** We implement 2 bits to perform BDR transformation.

**Grayscale.** We first calculate the weighted sum of the three channels and then replicate it across all three channels.

**Adversarial Training (AT).** PGD-10 is employed with a step size of $2/255$, training the model on CIFAR-10 for 100 epochs.

**Filters.** We use a kernel size of 3 for median, mean, and Gaussian smoothing (with a standard deviation of 0.1).

**Gaussian noise.** We generate noise for each sample with a distribution of $\mathcal{N}(0, 0.1)$.

**Cutout.** We adjust the sizes of the squared cutout box to match the image sizes of the datasets: 16 for CIFAR-10 and CIFAR-100, and 116 for the ImageNet subset. The cutout box is randomly placed within each image and maintains a consistent size across all images.

**Cutmix.** For a given sample, we first generate a square bounding box centered at a randomly chosen position with a randomly selected size (ranging from 0 to 1). The content within this bounding box is then replaced with content cropped from another randomly chosen image from the same mini-batch.

**Mixup.** We randomly select another image and blend it with the current sample using a randomly chosen weight (ranging from 0 to 1). We retain the original label for the current sample during loss function computation.

## A.2  Additional Experiment Results

### A.2.1  Ablation studies on different parameters.

In our experiments, we firstly test the performance when $\omega_1$, $\omega_2$, and $\omega_3$ are set as 0, which means we just focus exclusively on the hiding loss, denoted as $\mathcal{L}_{\text{hide}}$. The training loss trends for using only $\mathcal{L}_{\text{hide}}$ compared to our complete loss $\mathcal{L}_{\text{total}}$ are depicted in Figure A. The $\mathcal{L}_{\text{total}}$ exhibits a notable initial rise and subsequent decline, attributed to the optimization of the revealing loss. In contrast, when training with only the hiding loss, the training plot shows the loss remains 0. Since the minimal amount of information is hidden in the clean images during the first step, with the perturbation radius staying below the 8/255 threshold, thus fulfilling the optimization objectives without further optimization needed. Besides, we visualized the perturbations resulting from different designed losses. As Figure B shows, using only the hiding loss $\mathcal{L}_{\text{hide}}$ leads to minimal information being hidden in clean images and fails to achieve unlearnable performance. Furthermore, our evaluation of unlearnable examples generated by the model trained only with $\mathcal{L}_{hide}$ reveals that the test accuracy is close to that of clean images, as shown in Table. K. This indicates that minimal information

Table A: The experimental results of different settings on parameters of $\omega_1$.

| Setting | $\omega_1$ | Vanilla | Cutout | Cutmix | Mixup | MeanF | MedianF | BDR | Gray | GaussN | GaussF | JPEG10 | JPEG50 | AT | Mean |
|---|---|---|---|---|---|---|---|---|---|---|---|---|---|---|---|
| Ours(S) | 10 | 16.00 | 16.35 | 13.69 | 17.28 | 30.86 | 64.48 | 23.40 | 18.34 | 24.11 | 13.70 | 82.03 | 70.60 | 48.92 | 33.83 |
| | 1(ours) | 15.36 | 10.79 | 10.00 | 14.72 | 17.68 | 17.00 | 21.12 | 17.61 | 22.78 | 11.16 | 80.41 | 81.03 | 38.31 | **27.54** |
| | $10^{-1}$ | 13.54 | 17.32 | 14.11 | 17.24 | 29.90 | 37.61 | 28.54 | 19.15 | 25.25 | 16.23 | 81.48 | 63.96 | 58.30 | 32.51 |
| | $10^{-2}$ | 11.31 | 12.48 | 10.01 | 11.12 | 17.11 | 23.48 | 17.46 | 13.89 | 22.41 | 10.05 | 80.04 | 84.38 | 45.20 | 27.61 |
| Ours(C) | 10 | 11.60 | 10.10 | 12.52 | 10.83 | 11.92 | 21.24 | 17.48 | 11.45 | 12.42 | 10.00 | 76.42 | 20.26 | 17.56 | 18.75 |
| | 1(ours) | 10.00 | 10.00 | 11.25 | 10.02 | 10.59 | 10.04 | 13.53 | 10.00 | 10.00 | 10.00 | 72.97 | 23.62 | 10.00 | **16.31** |
| | $10^{-1}$ | 10.00 | 10.00 | 9.99 | 10.00 | 10.11 | 10.09 | 12.44 | 10.00 | 10.00 | 10.00 | 72.36 | 28.89 | 10.05 | 16.46 |
| | $10^{-2}$ | 10.00 | 10.00 | 10.02 | 10.00 | 10.09 | 10.00 | 16.34 | 10.00 | 10.00 | 10.00 | 72.45 | 58.50 | 10.00 | 19.03 |

is hidden in clean images, leading to ineffective unlearnability. Based on the above experimental results and our analyze, we apply our total loss $\mathcal{L}_{\text{total}}$ in all experiments.

In our parameters' ablation studies, we have conducted these experiments on CIFAR10 and tabulated the results, as shown in Table A B C. The optimal results were obtained when $\omega_1$ and $\omega_2$ were set to 1, and $\omega_3$ was set to 0.0001. This configuration is effective because the loss associated with $\omega_3$ is typically 1000 times larger than those of $\omega_1$ and $\omega_2$. Setting $\omega_3$ to 0.0001 helps balance the training process by ensuring that the scales of the losses remain consistent.

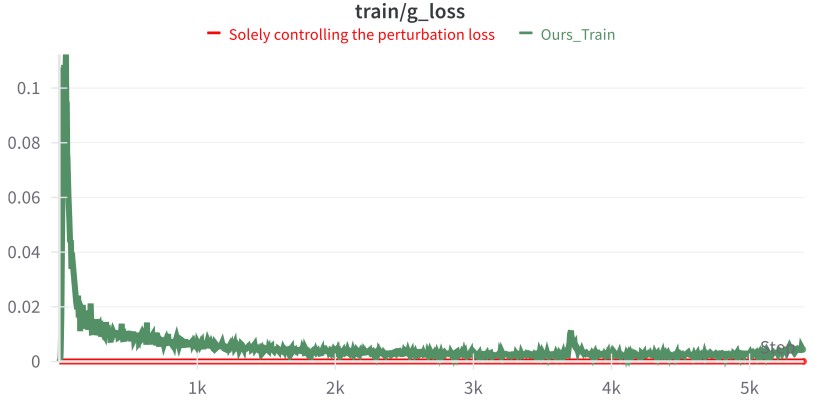

Figure A: Traning loss plogs with different loss designs.

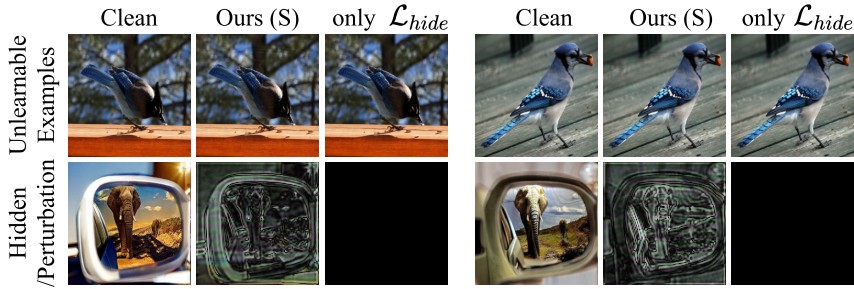

Figure B: Visualization of perturbation maps under different loss designs.

### A.2.2 Additional experiments on different hidden semantic image

To gain deeper insights into our design methodologies, additional ablation experiments were conducted, each focusing on the removal of a major component. These supplementary studies include: 1) employing clean images from the subsequent class as hidden images, and 2) utilizing natural semantic images from the CIFAR100 dataset as hidden images. 3) Moreover, to assess the impact of our CLIP-based clustering approach for text prompts, we implemented a variant using randomly generated prompts, thereby eliminating controlled inter-class differences. The results, as depicted in Table D, reveal some critical findings. In the class-wise setting, it appears that any form of semantic images can facilitate the generation of unlearnable examples. This indicates that the effectiveness

Table B: The experimental results of different settings on parameters of $\omega_2$.

| Setting | $\omega_2$ | Vanilla | Cutout | Cutmix | Mixup | MeanF | MedianF | BDR | Gray | GaussN | GaussF | JPEG10 | JPEG50 | AT | Mean |
|---|---|---|---|---|---|---|---|---|---|---|---|---|---|---|---|
| Ours(S) | 10 | 15.12 | 10.35 | 10.34 | 13.58 | 17.94 | 23.41 | 20.41 | 16.00 | 24.53 | 10.01 | 81.89 | 80.81 | 39.88 | 28.02 |
| | 1(ours) | 15.36 | 10.79 | 10.00 | 14.72 | 17.68 | 17.00 | 21.12 | 17.61 | 22.78 | 11.16 | 80.41 | 81.03 | 38.31 | **27.54** |
| | $10^{-1}$ | 16.99 | 16.81 | 11.01 | 12.22 | 25.72 | 35.44 | 26.33 | 20.33 | 27.34 | 16.66 | 82.40 | 81.74 | 69.43 | 34.03 |
| | $10^{-2}$ | 10.57 | 15.16 | 21.60 | 14.12 | 19.67 | 35.44 | 22.87 | 18.17 | 29.63 | 10.82 | 82.99 | 87.50 | 76.03 | 34.20 |
| Ours(C) | 10 | 10.00 | 10.00 | 11.22 | 11.94 | 10.09 | 10.10 | 13.27 | 10.00 | 10.00 | 10.08 | 77.23 | 22.81 | 10.12 | 16.68 |
| | 1(ours) | 10.00 | 10.00 | 11.25 | 10.02 | 10.59 | 10.04 | 13.53 | 10.00 | 10.00 | 10.00 | 72.97 | 23.62 | 10.00 | **16.31** |
| | $10^{-1}$ | 10..04 | 10.00 | 11.64 | 13.81 | 15.49 | 10.43 | 17.09 | 10.16 | 10.94 | 10.05 | 77.79 | 25.29 | 18.35 | 19.25 |
| | $10^{-2}$ | 10.00 | 10.00 | 10.03 | 10.28 | 10.41 | 10.36 | 15.29 | 10.87 | 10.00 | 10.00 | 79.46 | 29.51 | 10.00 | 17.40 |

Table C: The experimental results of different settings on parameters of $\omega_3$.

| Setting | $\omega_3$ | Vanilla | Cutout | Cutmix | Mixup | MeanF | MedianF | BDR | Gray | GaussN | GaussF | JPEG10 | JPEG50 | AT | Mean |
|---|---|---|---|---|---|---|---|---|---|---|---|---|---|---|---|
| Ours(S) | $10^{-2}$ | 14.09 | 13.71 | 10.06 | 11.50 | 17.65 | 28.47 | 22.44 | 17.63 | 22.67 | 15.79 | 82.04 | 76.56 | 39.96 | 28.66 |
| | $10^{-3}$ | 29.41 | 21.50 | 17.33 | 31.99 | 71.78 | 71.78 | 57.44 | 30.15 | 48.16 | 22.85 | 82.41 | 84.34 | 84.63 | 50.29 |
| | $10^{-4}$(ours) | 15.36 | 10.79 | 10.00 | 14.72 | 17.68 | 17.00 | 21.12 | 17.61 | 22.78 | 11.16 | 80.41 | 81.03 | 38.31 | **27.54** |
| | $10^{-5}$ | 11.10 | 14.35 | 13.08 | 19.27 | 23.38 | 40.61 | 47.25 | 28.73 | 53.54 | 10.62 | 83.18 | 89.32 | 83.69 | 39.86 |
| Ours(C) | $10^{-2}$ | 10.00 | 10.00 | 10.00 | 11.84 | 10.24 | 10.10 | 16.45 | 10.03 | 11.29 | 10.00 | 74.47 | 21.16 | 15.58 | 17.01 |
| | $10^{-3}$ | 10.00 | 10.01 | 10.11 | 10.65 | 12.13 | 19.81 | 15.06 | 15.69 | 10.21 | 10.00 | 77.98 | 21.37 | 10.77 | 17.98 |
| | $10^{-4}$(ours) | 10.00 | 10.00 | 11.25 | 10.02 | 10.59 | 10.04 | 13.53 | 10.00 | 10.00 | 10.00 | 72.97 | 23.62 | 10.00 | **16.31** |
| | $10^{-5}$ | 10.00 | 10.00 | 10.03 | 9.99 | 10.01 | 10.00 | 14.52 | 10.05 | 10.16 | 10.00 | 80.09 | 73.48 | 10.54 | 20.68 |

of unlearnability stems from the use of semantic images, highlighting robust generalizability across various countermeasures. For the sample-wise setting, our design demonstrates clear advantages. Without the inclusion of controlled inter-class differences – that is, simply using next-class images or random natural images – our proposed methods exhibit suboptimal performance. Similarly, employing random prompts without managing intra-class distances results in a minor decrease in test accuracy.

### A.2.3  ABLATION STUDIES ON DIFFERENT HIDING MODEL.

We have expanded our research to different image-hiding networks, notably the ISGAN (Zhang et al., 2019). Our experiments assess the effectiveness of ISGAN-hidden unlearnable examples, and the results as shown in Table H. The findings reveal that while unlearnability can be achieved with other deep hiding models like ISGAN, the performance is not as optimal as with our applied Invertible Neural Network (INN). INN demonstrates superior performance in deep hiding (Xu et al., 2022; Xiao et al., 2023), which is why we chose it as our baseline model to validate our concepts. We believe these distinctions, along with our comprehensive evaluations, underscore the unique contribution of our work in the field of image hiding and data privacy.

### A.2.4  TRANSFERABILITY OF THE PROPOSED METHOD

To further validate the effectiveness of the methods proposed in this paper, we conducted more comprehensive cross-validation. 1. We verify the robustness of our UEs against various countermeasures under different architectures, and the results as shown in Table E. It supports our claim that the proposed deep hiding UEs maintain their efficacy against different countermeasures across architectures. 2. We conducted the transferability study across architectures with limited unlearnable examples, and the results as shown in Table F. The test accuracy decreases in a similar trend when we increase the percentage of the unlearnable examples.

### A.2.5  PERFORMANCE ON DIFFERENT UNLEARNABLE PERCENTAGES

We further evaluate the performance of our proposed approach by using varying mixtures of clean images and unlearnable examples, the results as shown in Table G.

### A.2.6  PERFORMANCE ON GENERATED UNLEARNABLE EXAMPLES AGAINST JPEG COMPRESSION.

We further explore the lower JPEG values, the results as shown in Table I. We found that lower JPEG values result in even lower test accuracy, which confirms that stronger compression not only damages the unlearnable example perturbations but also distorts the original image features significantly. We also show examples of images compressed by different JPEG compression.

Table D: Table D: Additional experimental results on CIFAR10 by using different hidden semantic images, including images from the next class, random natural images (CIFAR100), and our semantic image generation module.

| Setting | Semantic Images | Vanilla | Cutout | Cutmix | Mixup | MeanF | MedianF | BDR | Gray | GaussN | GaussF | JPEG10 | JPEG50 | AT | Mean |
|---|---|---|---|---|---|---|---|---|---|---|---|---|---|---|---|
| Ours(C) | Next Class | 10.00 | 10.00 | 9.96 | 10.18 | 10.47 | 10.18 | 10.00 | 10.00 | 10.00 | 10.00 | 62.91 | 17.26 | 10.01 | 14.69 |
| | Random Natural Images | 10.00 | 10.00 | 11.41 | 10.00 | 10.01 | 11.71 | 10.00 | 10.01 | 10.00 | 10.00 | 58.38 | 10.05 | 10.00 | 13.97 |
| | Random Prompt | 10.00 | 10.00 | 10.00 | 12.32 | 9.72 | 10.00 | 10.00 | 10.59 | 10.00 | 10.00 | 71.44 | 30.27 | 10.00 | 16.49 |
| | Ours | 10.00 | 10.00 | 11.25 | 10.02 | 10.59 | 10.04 | 13.53 | 10.00 | 10.00 | 10.00 | 72.97 | 23.62 | 10.00 | 16.31 |
| Ours(S) | Next Class | 80.19 | 75.79 | 74.79 | 71.26 | 60.57 | 82.25 | 72.89 | 67.46 | 72.38 | 78.66 | 82.01 | 87.68 | 82.53 | 76.04 |
| | Random Natural Images | 94.09 | 94.34 | 93.84 | 94.17 | 64.55 | 85.94 | 88.86 | 91.81 | 88.70 | 94.20 | 82.84 | 90.31 | 85.31 | 88.38 |
| | Random Prompt | 10.00 | 16.53 | 20.81 | 17.14 | 18.51 | 21.73 | 24.98 | 13.85 | 22.07 | 10.59 | 80.09 | 82.90 | 46.54 | 29.67 |
| | Ours | 15.36 | 10.79 | 10.00 | 14.72 | 17.68 | 17.00 | 21.12 | 17.61 | 22.78 | 11.16 | 80.41 | 81.03 | 38.31 | 27.54 |

Table E: Test accuracy (%) of model train on unlearnable examples from CIFAR-10 with five architectures, including ResNet-18 (R18), ResNet-50 (R50), VGG-19 (V19), and DenseNet-121 (D121), and Vision Transformer (ViT), against data augmentations, data preprocessing, and adversarial training.

| Settings | Model | Vanilla | Cutout | Cutmix | Mixup | MeanF | MedianF | BDR | Gray | GaussN | GaussF | JPEG10 | JPEG50 | AT | Mean |
|---|---|---|---|---|---|---|---|---|---|---|---|---|---|---|---|
| Ours(S) | R18 | 15.36 | 10.79 | 10.00 | 14.72 | 17.68 | 17.00 | 21.12 | 17.61 | 22.78 | 11.16 | 80.41 | 81.03 | 38.31 | 27.54 |
| | R50 | 13.32 | 12.42 | 12.99 | 11.60 | 18.70 | 22.70 | 23.77 | 12.62 | 17.24 | 16.23 | 80.70 | 78.30 | 37.11 | 27.52 |
| | V19 | 10.45 | 17.25 | 14.37 | 17.87 | 23.62 | 32.27 | 22.77 | 17.72 | 22.28 | 14.89 | 80.61 | 80.64 | 52.55 | 31.33 |
| | D121 | 18.88 | 21.16 | 12.52 | 18.81 | 53.41 | 53.10 | 28.10 | 18.06 | 12.54 | 18.22 | 77.93 | 76.46 | 70.58 | 36.91 |
| | ViT | 15.80 | 20.64 | 21.93 | 10.67 | 54.05 | 54.57 | 25.40 | 15.20 | 22.45 | 21.38 | 65.38 | 64.91 | 49.50 | 33.99 |
| Ours(C) | R18 | 10.00 | 10.00 | 11.25 | 10.02 | 10.59 | 10.04 | 13.53 | 10.00 | 10.00 | 10.00 | 72.97 | 23.62 | 10.00 | 16.31 |
| | R50 | 10.00 | 10.00 | 11.02 | 10.04 | 10.06 | 10.45 | 17.31 | 10.00 | 10.00 | 10.00 | 74.43 | 24.15 | 11.09 | 16.81 |
| | V19 | 10.58 | 10.17 | 10.02 | 17.46 | 10.78 | 12.86 | 15.79 | 10.30 | 10.03 | 10.03 | 71.77 | 24.84 | 13.44 | 17.54 |
| | D121 | 10.00 | 10.00 | 10.60 | 10.23 | 10.49 | 10.01 | 11.63 | 10.00 | 10.00 | 10.53 | 72.85 | 22.06 | 10.00 | 16.03 |
| | ViT | 10.00 | 10.01 | 10.02 | 10.67 | 11.23 | 22.34 | 12.35 | 10.12 | 10.00 | 10.00 | 61.92 | 30.69 | 18.79 | 17.55 |

### A.2.7 EVALUATION OF INTER-CLASS AND INTRA-CLASS DISTANCES

We investigate the intra-/inter-class distance in latent features using a trained (for unlearnable examples) or pre-trained (for clean images) ResNet18. We employ the output of the last CNN layer as the latent feature for each sample and average these to determine the mean latent feature per class. For intra-class distance, we compute and report the mean cosine similarity between each sample's latent feature and its class mean. For inter-class distance, we calculate the mean cosine similarity between each class's mean latent feature and the overall dataset mean.

We present our findings in Table J. It is noteworthy that our unlearnable examples, in both class-wise and sample-wise settings, exhibit significantly reduced intra-class distances, as evidenced by a higher cosine similarity approaching 1. Furthermore, by compacting the semantics within each class, we also achieve an increased inter-class distance. These outcomes suggest that our method successfully generates unlearnable examples characterized by minimal intra-class distances and maximized inter-class distances, thereby enhancing unlearnability.

| Clean Images | Unlearnable Examples | JPEG2 | JPEG4 | JPEG6 | JPEG8 | JPEG10 | JPEG50 |
|---|---|---|---|---|---|---|---|

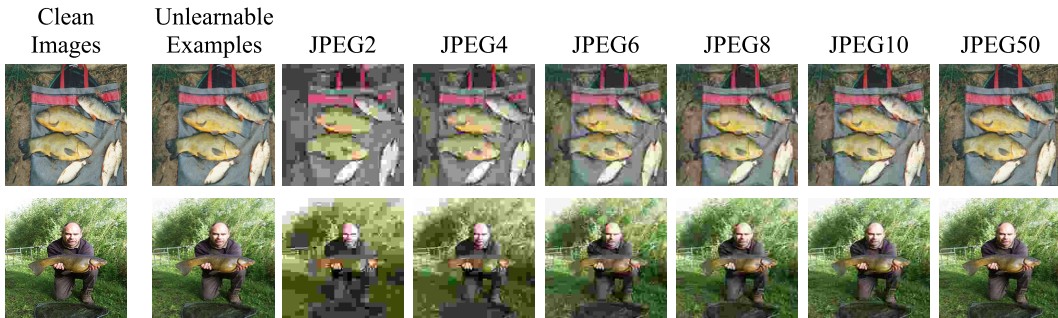

Figure C: Visualization of Clean and Unlearnable Images under Varied JPEG Compression.

### A.2.8 GENERATED SEMANTIC IMAGES

We display several generated semantic images for two groups in Figure D and Figure E. Each group of semantic images, shares similar semantic features but differ in textures, colors, and other low-level features. Besides, Figure F provides more examples of different semantic images generated for different classes.

Table F: Test accuracy (%) of CIFAR-10 on the different models trained by the clean data mixed with different percentages of unlearnable examples.

| Setting | Model | 20% | 40% | 60% | 80% |
|---|---|---|---|---|---|
| Ours(S) | R18 | 93.73 | 92.41 | 90.08 | 84.40 |
| | R50 | 94.16 | 92.82 | 90.51 | 85.66 |
| | V19 | 92.11 | 91.14 | 88.77 | 83.48 |
| | D121 | 89.02 | 87.77 | 84.94 | 81.39 |
| | ViT | 75.95 | 74.77 | 74.09 | 70.50 |
| Ours(C) | R18 | 93.53 | 92.67 | 89.99 | 84.47 |
| | R50 | 94.04 | 92.31 | 90.58 | 85.15 |
| | V19 | 92.13 | 90.23 | 87.99 | 80.60 |
| | D121 | 88.28 | 86.56 | 83.57 | 77.79 |
| | ViT | 75.44 | 74.58 | 69.34 | 64.80 |

Table G: Test accuracy (%) of CIFAR-10 on the models trained by the clean data mixed with different percentages of unlearnable examples.

| Percentage (%) | 20 | 40 | 60 | 80 | 85 | 90 | 92 | 94 | 96 | 98 | 100 |
|---|---|---|---|---|---|---|---|---|---|---|---|
| Ours(S) | 93.73 | 92.41 | 90.08 | 84.40 | 84.07 | 81.37 | 79.92 | 77.30 | 71.93 | 59.34 | 10.00 |
| Ours(C) | 93.53 | 92.67 | 89.99 | 84.47 | 84.20 | 81.41 | 78.42 | 75.71 | 66.90 | 53.00 | 15.36 |

### A.2.9 BYPRODUCT OF LOGO HIDING AND EXTRACTION

Benefiting from the invertible performance of the INN model itself, our method is able to recover hidden semantic images from unlearnable examples. Hence, we further demonstrate a byproduct of logo hiding and extraction for source tracing. As illustrated in Figure G, we show that our Deep Hiding scheme could not only achieve unlearnability for data protection but also hide and reveal a predefined logo for data source tracing.

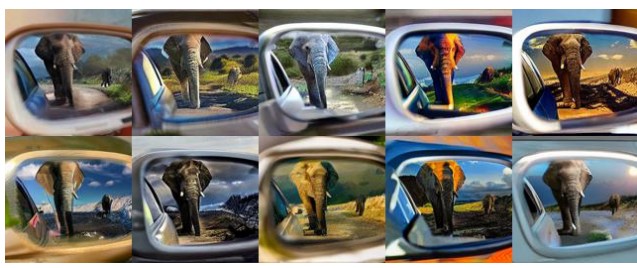

Figure D: Visualization of ten images of generated semantic images for one class.

Table H: Test accuracy (%) of model train on unlearnable examples generated by using another deep hiding model (ISGAN).

| ISGAN | Vanilla | Cutout | Cutmix | Mixup | MeanF | MedianF | BDR | Gray | GaussN | GaussF | JPEG10 | JPEG50 | AT | Mean |
|---|---|---|---|---|---|---|---|---|---|---|---|---|---|---|
| Ours(S) | 54.05 | 53.74 | 55.97 | 72.61 | 48.83 | 85.11 | 86.77 | 53.04 | 88.45 | 54.25 | 84.35 | 90.55 | 87.22 | 70.38 |
| Ours(C) | 23.99 | 24.07 | 28.42 | 42.69 | 37.56 | 79.3 | 83.64 | 30.06 | 87.92 | 25.24 | 84.46 | 90.28 | 87.46 | 55.78 |

Table I: Test accuracy (%) of model train on unlearnable examples from CIFAR10 against JPEG compression.

| JPEG Quality Factor | 2 | 4 | 6 | 8 | 10 | 50 |
|---|---|---|---|---|---|---|
| Ours(S) | 68.01 | 74.35 | 77.39 | 78.56 | 80.41 | 81.03 |
| Ours(C) | 64.53 | 70.18 | 72.56 | 72.72 | 72.97 | 23.62 |

Table J: Evaluation of Inter-Class and Intra-Class Distances.

| Data | Intra-class | Inter-class |
|---|---|---|
| Clean images | 0.8457 | 0.8742 |
| Ours(S) | 0.9702 | 0.9030 |
| Ours(C) | 0.9882 | 0.8815 |

Table K: Evaluation on the different hiding model trained by solely controlling the hiding loss ($\mathcal{L}_{\text{hide}}$) and using our designed loss ($\mathcal{L}_{\text{total}}$). The test accuracy (%) are evaluated on CIFAR-10 in the class-wise setting.

| Method | Vanilla | Cutout | Cutmix | Mixup | MeanF | MedianF | BDR | Gray | GaussN | GaussF | JPEG10 | JPEG50 | AT | Mean |
|---|---|---|---|---|---|---|---|---|---|---|---|---|---|---|
| Clean | 94.59 | 95.00 | 94.77 | 94.96 | 49.70 | 86.64 | 89.07 | 92.80 | 88.71 | 94.54 | 85.22 | 90.89 | 84.19 | 87.78 |
| only $\mathcal{L}_{\text{hide}}$ | 94.71 | 95.05 | 94.68 | 95.4 | 38.88 | 87.39 | 89.12 | 93.1 | 88.55 | 87.39 | 85.04 | 91.04 | 88.59 | 86.84 |
| ours | 10.00 | 10.00 | 11.25 | 10.02 | 10.59 | 10.04 | 13.53 | 10.00 | 10.00 | 10.00 | 72.97 | 23.62 | 10.00 | 16.31 |

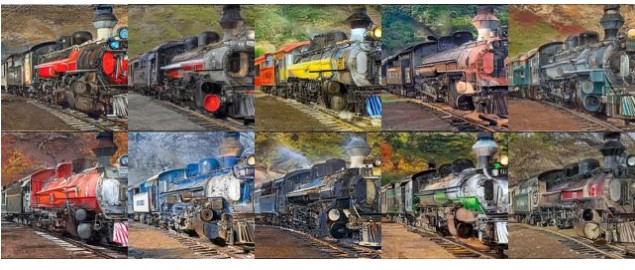

Figure E: Visualization of ten images of generated semantic images for one class.

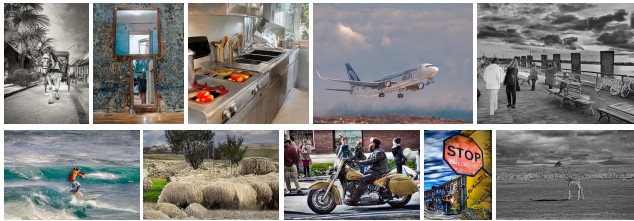

Figure F: Visualization of generated semantic images for ten different classes.

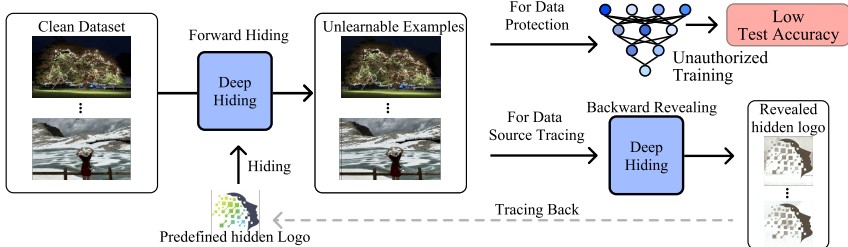

Figure G: Overview of byproduct. Our proposed DH scheme could hide a predefined logo into a clean dataset via the forward hiding process to generate unlearnable examples for data protection, and then extract the hidden logo from the unlearnable examples for source tracing.