# OpenReview forum: "Towards robust unlearnable examples via deep hiding"
_ICLR.cc/2024/Conference — Submitted to ICLR 2024_

### Official Review · Reviewer_Xq8x · 2023-10-19

**Soundness:** 3 good
**Presentation:** 3 good
**Contribution:** 2 fair
**Rating:** 5
**Confidence:** 5

**Summary:**

The majority of existing methods for generating Unlearnable Examples primarily focus on investigating the robustness against adversarial training, while overlooking the resilience to other attack strategies such as data augmentation and preprocessing. Previous research has found that images containing semantic information are robust against common attacks. In this paper, the authors propose a novel defense mechanism that is effective under various prevalent attack methods. The authors first generate multiple image samples corresponding to the number of categories using ControlNet, and then employ Image Hiding techniques to conceal the generated image samples within the dataset to be protected, utilizing Invertible Neural Networks (INN). This process disrupts the original semantic information of the images, thereby achieving the goal of protecting the data.

**Strengths:**

- The authors introduce Image Hiding techniques into Data Unlearning, providing a new reference direction for the field of Data Unlearning.
- The authors employ a Latent Feature Concentration module during the image hiding process to achieve consistency in semantic features within the same class, thus allowing the features of similar data to become more concentrated.
- The selection of attack methods for the experiments in this paper is fairly comprehensive.

**Weaknesses:**

- The method consists of three modules: The Deep Hiding Scheme, which introduces the concept of Image Hiding, an existing work; the Semantic Image Generation Module, which utilizes the existing ControlNet; and the Latent Feature Concentration Module, which is very similar to the idea of EntF mentioned in the related work. In summary, the ideas presented in this paper are intriguing, but the innovation is insufficient.
- Previous research has shown that some Unlearnable Examples possess an inherent resistance to data augmentation, which contradicts the paper's claim that prior methods have overlooked these issues.
- In the experimental settings of the paper, the perturbation for adversarial training is set to $8/255$, and the perturbation for protection noise also appears to be set to $8/255$ according to the experimental table. Under this setting, which is consistent with EntF, the paper's experiments lack a comparison with this method.
- The paper doesn't explain why it's also effective under adversarial training.

Although the introduction of Image hiding in the paper is interesting, the three important parts of the article are existing work and lack innovation. In addition, there is a lack of comparison of EntF methods, a lack of inquiry about defense against training, and a lack of inquiry about the effect of the generated hiding image on protection. All in all, at this point in time, I would recommend this paper as weak reject.

**Questions:**

see weaknesses

---

> ### Author Response · Authors · 2023-11-20
>
> **Q1:** The method consists of three modules: The Deep Hiding Scheme, which introduces the concept of Image Hiding, an existing work; the Semantic Image Generation Module, which utilizes the existing ControlNet; and the Latent Feature Concentration Module, which is very similar to the idea of EntF mentioned in the related work. In summary, the ideas presented in this paper are intriguing, but the innovation is insufficient.
>
> **Ans:** Thank you for the comments. We would like to clarify that our Latent Feature Concentration module (LFC) differs from EntF in various aspects. Firstly, EntF aims to generate entangled features associated with perturbed data, causing confusion in models; while our module encourages the concentration of the perturbation itself, creating stronger shortcuts. Secondly, EntF focuses on robustness against adversarial training, while we use LFC to achieve more general robustness. Besides, we would like to emphasize our novelty in semantic-based unlearnable examples. The proposed modules (i.e., Semantic Image Generation (SIG) and LFC) all combine to increase the inter-class distance and reduce the intra-class variance, enhancing the general robustness of unlearnable examples, shown in Table 4.
>
>
>
>
> **Table 4: Ablation studies on CIFAR10 for designed Latent Feature Concentration module (LFC), and Semantic Images Generation module (SIG), including Text Prompts Clustering (TPC) and Stable Diffusion model and ControlNet (SD+C).**
>
>
> | Setting | LFC | SIG: TPC | SIG: SD+C | Vanilla | Cutout | Cutmix | Mixup | MeanF | MedianF | BDR   | Gray  | GaussN | GaussF | JPEG10 | JPEG50 | AT    | Mean  |
> |:-------:|:---:|:--------:|:---------:|:-------:|:------:|:------:|:-----:|:-----:|:-------:|:-----:|:-----:|:------:|:------:|:------:|:------:|:-----:|:-----:|
> | Sample-wise|   ×  |       ×   |        ×   | 94.08   | 94.52  | 94.07  | 88.23 | 65.42 | 87.04   | 89.46 | 91.45 | 88.36  | 94.07  | 83.29  | 89.97  | 86.26 | 88.17 |
> |    Sample-wise     | ×    | ✓        | ✓         | 14.77   | 22.20  | 13.06  | 23.44 | 30.18 | 51.43   | 35.66 | 17.31 | 37.50  | 15.80  | 81.97  | 81.48  | 81.24 | 38.93 |
> |  Sample-wise       | ✓   |    ×       | ✓         | 10.00   | 16.53  | 20.81  | 17.14 | 18.51 | 21.73   | 24.98 | 13.85 | 22.07  | 10.59  | 80.09  | 82.90  | 46.54 | 29.67 |
> |    Sample-wise     | ✓   |    ×       |    ×        | 94.09   | 94.34  | 93.84  | 94.00 | 64.55 | 85.94   | 88.86 | 91.00 | 88.70  | 94.20  | 82.84  | 90.31  | 85.31 | 88.38 |
> |     Sample-wise    | ✓   | ✓        | ✓         | 15.36   | 10.79  | 10.00  | 14.72 | 17.68 | 17.00   | 21.12 | 17.61 | 22.78  | 11.16  | 80.41  | 81.03  | 38.31 | **27.54** |
> | Class-wise |  ×    |     ×      |    ×        | 12.16   | 11.02  | 11.43  | 10.81 | 10.14 | 13.28   | 10.10 | 12.12 | 10.00  | 10.00  | 78.21  | 15.32  | 12.87 | 16.73 |
> |    Class-wise     |  ×    | ✓        | ✓         | 10.00   | 10.00  | 10.06  | 10.82 | 13.26 | 25.76   | 15.83 | 14.20 | 10.03  | 10.00  | 76.26  | 20.28  | 14.78 | 18.56 |
> |    Class-wise     | ✓   |      ×     | ✓         | 10.00   | 10.00  | 10.00  | 12.32 | 9.72  | 10.00   | 10.00 | 10.59 | 10.00  | 10.00  | 71.44  | 30.27  | 10.00 | 16.49 |
> |    Class-wise     | ✓   |     ×      |    ×        | 10.00   | 10.00  | 11.41  | 10.00 | 10.01 | 11.71   | 10.00 | 10.01 | 10.00  | 10.00  | 58.38  | 10.05  | 10.00 | **13.97** |
> |    Class-wise     | ✓   | ✓        | ✓         | 10.00   | 10.00  | 11.25  | 10.02 | 10.59 | 10.04   | 13.53 | 10.00 | 10.00  | 10.00  | 72.97  | 23.62  | 10.00 | 16.31 |
>
> **Q2:** Previous research has shown that some Unlearnable Examples possess inherent resistance to data augmentation, which contradicts the paper's claim that prior methods have overlooked these issues.
>
> **Ans:** We agree that previous works have shown resistance to data augmentations to some extent. However, the recent work ISS[G] found that these methods are extremely vulnerable to simple data processing like Grayscaling and JPEG compression with low quality factors. We follow the protocol in ISS and gain insights from the results in Table 1:
> - The existing methods are not generally robust against different types of data augmentations and processing. For example, though EM and REM are resistant to popular data augmentations like cutout and mixup, they are very vulnerable to grayscaling and JPEG compression.
> - The existing methods are not resistant to the countermeasures with higher-level severity. We find most of the existing methods will be mitigated once the quality factor of JPEG decreases to 10.
> As a result, we argue that the resistance of UEs against countermeasures is not generally solved. We aim to improve general robustness with the deep hiding technique in this paper.
>
> ```
> [G] Zhuoran Liu, Zhengyu Zhao, and Martha A. Larson. "Image Shortcut Squeezing: Countering Perturbative Availability Poisons with Compression." In International Conference on Machine Learning, 2023.
> ```

---

> > ### Author Response · Authors · 2023-11-20
> >
> > **Q3:** In the experimental settings of the paper, the perturbation for adversarial training is set to 8/255, and the perturbation for protection noise also appears to be set to 8/255 according to the experimental table. Under this setting, which is consistent with EntF, the paper's experiments lack a comparison with this method.
> >
> > **Ans:** Thank you for suggesting a comparison with EntF. EntF mainly focuses on robustness against adversarial training, and its performance on other countermeasures is relatively poor. Hence, we didn't include its comparison in the initial paper. In the modified version, we have included the results of EntF on CIFAR10, CIFAR100 in Table 1. Besides, we study its transferability across architectures and its effectiveness in clean-poison mixed settings. The results show that EntF remains robust against adversarial training, demonstrating competitive performance among the surrogate-dependent UEs including EM, REM, TAP. However, its general robustness is not as good as other methods.
> >
> > **Q4:** The paper doesn't explain why it's also effective under adversarial training.
> >
> >
> > **Ans:** Our proposed method hides semantic-based perturbation, instead of gradient-based perturbation. We hypothesize they fall in different latent spaces, so our perturbation cannot be erased by adversarial training, similar to OPS [H]. We added the explanation in Section 4.3.
> >
> > ```
> > [H] Wu S, Chen S, Xie C, et al. One-pixel shortcut: on the learning preference of deep neural networks. ICLR, 2023.
> > ```
> >
> > **Q5(summary):** Although the introduction of Image hiding in the paper is interesting, the three important parts of the article are existing work and lack innovation. In addition, there is a lack of comparison of EntF methods, a lack of inquiry about defense against training, and a lack of inquiry about the effect of the generated hiding image on protection.
> >
> > **Ans:** Thank you for your summary. We respond to the first 3 points in the previous questions. For the inquiry about the effect of generated hiding images, we have conducted more comprehensive ablation studies to show the effect of each component. We show the complete result in the modified Table 4 and Table D in the supplementary. The results show that each component can contribute to the unlearnability. We hope the response can address your concerns well.

---

> > > ### Author Response · Authors · 2023-11-20
> > >
> > > **Table 1: Test accuracy (\%) of models trained on unlearnable examples from CIFAR-10, CIFAR-100, and ImageNet subset against data augmentations, data preprocessing, and adversarial training.**
> > >
> > >
> > > |                 | Method  | Vanilla | Cutout    | Cutmix | Mixup | MeanF   | MedianF | BDR   | Gray  | GaussN | GaussF  | JPEG10 | JPEG50 | AT      | Mean  |
> > > |:---------------:|:-------:|:-------:|:---------:|:------:|:-----:|:-------:|:-------:|:-----:|:-----:|:------:|:-------:|:------:|:------:|:-------:|:-----:|
> > > | CIFAR-10        | Clean   | 94.59   | 95.00     | 94.77  | 94.96 | 49.70   | 86.64   | 89.07 | 92.80 | 88.71  | 94.54   | 85.22  | 90.89  | 84.19   | 87.78 |
> > > |  CIFAR-10    | EM      | 10.00 | 10.00 | 15.39  | 16.82 | *10.63* | 24.27   | 35.90 | 69.29 | 32.96  | *10.01* | 84.80  | 87.82  | 84.28   | 37.87 |
> > > | CIFAR-10  | REM     | 29.00   | 29.42     | 26.13  | 28.37 | 19.07   | 32.80   | 39.93 | 69.83 | 39.97  | 28.67   | 84.15  | 77.65  | 85.93   | 45.46 |
> > > | CIFAR-10 | TAP     | 25.90   | 32.69     | 26.77  | 40.46 | 31.68   | 65.12   | 80.25 | 26.36 | 88.66  | 26.09   | 84.77  | 90.31  | 83.57   | 56.39 |
> > > | CIFAR-10  | **EntF**    | 91.50   | 91.30     | 90.93  | 92.52 | 17.85   | 70.28   | 91.46 | 80.33 | 90.31  | 79.79   | 74.36  | 83.56  | 75.86   | 79.23 |
> > > | CIFAR-10  | LSP     | 19.07   | 19.87     | 20.89  | 26.99 | 28.85   | 29.85   | 66.19 | 82.47 | 19.25  | 16.19   | 83.01  | 57.87  | 84.59   | 42.70 |
> > > |CIFAR-10  | AR      | 13.31   | 11.35     | 12.21  | 13.30 | 12.38   | 17.04   | 37.42 | 34.81 | 42.29  | 12.56   | 85.08  | 89.63  | 58.23   | 33.82 |
> > > | CIFAR-10 | OPS     | 16.53   | 89.73     | 83.91  | 34.88 | 17.31   | 86.86   | 43.04 | 16.65 | 36.72  | 15.10   | 82.79  | 57.00  | 9.42    | 45.38 |
> > > | CIFAR-10 | Ours(S) | 15.36   | 10.79     | 10.00  | 14.72 | 17.68   | 17.00   | 21.12 | 17.61 | 22.78  | 11.16   | 80.41  | 81.03  | 38.31   | 27.54 |
> > > | CIFAR-10 | Ours(C) | 10.00   | 10.00     | 11.25  | 10.02 | 10.59   | 10.04   | 13.53 | 10.00 | 10.00  | 10.00   | 72.97  | 23.62  | 10.00   | 16.31 |
> > > | CIFAR-100       | Clean   | 75.82   | 74.45     | 76.32  | 77.07 | 14.72   | 50.72   | 63.51 | 70.04 | 62.41  | 75.86   | 57.35  | 68.59  | 58.25   | 62.44 |
> > > | CIFAR-100  | EM      | 2.84    | 12.05     | 7.67   | 12.86 | 13.52   | 43.61   | 62.12 | 62.37 | 62.01  | 73.47   | 57.29  | 67.50  | 57.89   | 41.17 |
> > > |CIFAR-100   | REM     | 7.13    | 10.32     | 11.25  | 8.65  | 5.90    | 12.31   | 19.95 | 48.48 | 26.27  | 7.32    | 57.15  | 65.10  | 58.9    | 26.06 |
> > > |CIFAR-100  | TAP     | 14.00   | 16.55     | 15.99  | 22.56 | 5.86    | 31.95   | 55.12 | 8.90  | 61.4   | 13.95   | 56.56  | 66.67  | 56.53   | 32.77 |
> > > |CIFAR-100   | **EntF**   | 72.55   | 69.65     | 70.68  | 73.81 | 8.67    | 36.87   | 55.22 | 67.00 | 58.54  | 73.10   | 51.42  | 63.69  | 52.44   | 57.97 |
> > > | CIFAR-100   | LSP     | 2.68    | 2.55      | 2.69   | 4.39  | 7.15    | 6.76    | 28.23 | 42.77 | 22.42  | 2.19    | 55.23  | 33.60  | 57.45   | 20.62 |
> > > | CIFAR-100   | AR      | 1.50    | 1.47      | 1.56   | 1.37  | 5.35    | 3.89    | 28.28 | 19.68 | 59.34  | 1.57    | 56.99  | 65.72  | 58.33   | 23.47 |
> > > | CIFAR-100   | OPS     | 11.69   | 71.36     | 64.25  | 12.59 | 3.18    | 49.74   | 19.31 | 18.70 | 17.30  | 11.79   | 56.72  | 48.71  | 10.22   | 30.43 |
> > > |CIFAR-100   | Ours(S) | 4.79    | 4.13      | 5.39   | 4.72  | 6.22    | 10.21   | 12.12 | 3.72  | 19.85  | 3.61    | 49.50  | 34.86  | 41.12   | 15.40 |
> > > | CIFAR-100  | Ours(C) | 1.47    | 1.03      | 1.06   | 1.47  | 1.04    | 1.45    | 1.72  | 1.38  | 1.08   | 1.00    | 44.58  | 25.45  | 1.39    | 6.47  |
> > > | ImageNet subset | Clean   | 63.93   | 64.02     | 55.10  | 64.55 | 19.92   | 36.08   | 56.63 | 68.35 | 50.62  | 65.40   | 56.83  | 69.36  | 48.24   | 55.31 |
> > > |ImageNet subset | EM      | 28.99   | 18.78     | 17.61  | 36.55 | 7.46    | 32.60   | 53.43 | 17.93 | 44.63  | 26.04   | 53.41  | 56.96  | *43.56* | 33.69 |
> > > |ImageNet subset| REM     | 14.78   | 14.10     | 11.73  | 19.88 | 15.32   | 14.12   | 16.48 | 44.74 | 15.96  | 15.34   | 50.50  | 17.14  | 47.52   | 22.89 |
> > > | ImageNet subset | TAP     | 7.96    | 15.02     | 15.18  | 23.08 | 10.44   | 15.02   | 47.97 | 22.93 | 46.84  | 12.80   | 53.40  | 37.98  | 44.18   | 27.14 |
> > > |ImageNet subset | LSP     | 18.18   | 9.52      | 34.16  | 9.76  | 4.14    | 5.20    | 43.38 | 52.66 | 34.28  | 17.92   | 51.80  | 49.06  | 42.26}  | 28.64 |
> > > |ImageNet subset | Ours(S) | 3.36    | 2.14      | 3.30   | 2.32  | 2.52    | 6.48    | 8.62  | 1.32  | 4.02   | 1.94    | 39.92  | 26.66  | 44.56   | 11.32 |
> > > | ImageNet subset | Ours(C) | 1.02    | 1.56      | 1.28   | 1.44  | 1.14    | 1.74    | 2.94  | 1.32  | 1.70   | 0.98    | 34.78  | 12.00  | 44.00   | 8.15  |

---

> > > > ### Author Response · Authors · 2023-11-20
> > > >
> > > > **Table 2: Test accuracy (\%) of CIFAR-10 and CIFAR-100 on five architectures, including ResNet-18 (R18), ResNet-50 (R50), VGG-19 (V19), and DenseNet-121 (D121), and Vision Transformer (ViT).**
> > > > | Model      | R18-CIFAR10 | R50-CIFAR10 | V19-CIFAR10 | D121-CIFAR10 | ViT-CIFAR10 | R18-CIFAR100| R50-CIFAR100| V19-CIFAR100 | D121-CIFAR100 | ViT-CIFAR100|
> > > > |:----------:|:-----------------:|:-----------------:|:-----------------------------:|:-----------------:|:-----------------:|:----------------:|:----------------:|:----------------:|:----------------:|:----------------:|
> > > > | EM         | 10.10 | 10.00      | 10.82             | 12.56 | 11.88 | 2.84             | 3.88             | 9.23             | 64.87            | 7.65             |
> > > > | REM        | 30.40             | 25.10             | 24.54                         | 30.28             | 32.36             | 7.13             | 7.45             | 5.26             | 12.47            | 6.91             |
> > > > | TAP        | 25.93             | 25.48             | 30.36                         | 78.59             | 70.96             | 14.00            | 14.25            | 33.18            | 52.64            | 14.49            |
> > > > | **EntF** | 91.50             | 91.83             | 88.17                         | 83.30             | 69.23             | 72.55            | 73.19            | 65.68            | 60.85            | 49.43            |
> > > > | LSP        | 16.99             | 14.55             | 11.53                       | 24.83             | 23.78             | 2.68             | 4.06             | 2.84 | 27.05            | 9.40             |
> > > > | OPS        | 17.46             | 16.73             | 19.12                         | 18.40             | 28.09             | 11.69            | 10.90            | 5.67             | 10.38            | 17.23            |
> > > > | AR         | 11.88             | 15.83             | 13.21                         | 22.28             | 19.84             | 1.50 | 2.13 | 3.48             | 19.55            | 5.69 |
> > > > | Ours(S)    | 15.36           | 12.66 | {13.16}                       | 16.81             | 17.83             | 4.79             | 4.90             | 6.00             | 7.02 | 10.10            |
> > > > | Ours(C)    | 10.00      | 10.00    | 10.00                | 10.20    | 10.01    | 1.47    | 1.00    | 0.95    | 1.00    | 1.38    |
> > > >
> > > >
> > > >
> > > >
> > > > **Table 3: Test accuracy (\%) of CIFAR-10 on the models trained by the clean data mixed with different percentages of unlearnable examples.**
> > > >
> > > >
> > > > | Method     | 20% | 40% | 60% | 80% |
> > > > |:----------:|:------------------------:|:------------------------:|:------------------------:|:------------------------:|
> > > > | EM         | 94.30| 93.09| 91.42                    | 87.29                    |
> > > > | REM        | 93.83| 92.69| 91.12                    | 86.92                    |
> > > > | TAP        | 93.82| 92.78| 91.96                    | 88.49                    |
> > > > | **EntF** | 93.40| 91.71| 91.25                    | 91.07                    |
> > > > | LSP        | 93.50| 92.47        | 90.21                    | 84.81                    |
> > > > | OPS        | 93.64                    | 92.63                    | 90.05        | 84.42        |
> > > > | AR         | 94.07| 92.66                    | 90.34                    | 85.18                    |
> > > > | Ours(S)    |93.53| 92.67| 89.99             | 84.47                    |
> > > > | Ours(C)    | 93.73| 92.41| 90.08| 84.40
> > > >
> > > > **Table D: Additional experimental results on CIFAR10 by using different hidden semantic images, including images from the next class, random natural images (CIFAR100), and our semantic image generation module.**
> > > >
> > > > | Setting| Semantic Images| Vanilla | Cutout | Cutmix | Mixup | MeanF | MedianF | BDR   | Gray  | GaussN | GaussF | JPEG10 | JPEG50 | AT    | Mean  |
> > > > |:---------------:|:---------------------:|:-------:|:------:|:------:|:-----:|:-----:|:-------:|:-----:|:-----:|:------:|:------:|:------:|:------:|:-----:|:-----:|
> > > > | Sample-wise | Next Class| 80.19   | 75.79  | 74.79  | 71.26 | 60.57 | 82.25   | 72.89 | 67.46 | 72.38  | 78.66  | 82.01  | 87.68  | 82.53 | 76.04 |
> > > > |Sample-wise| Random Natural Images | 94.09   | 94.34  | 93.84  | 94.17 | 64.55 | 85.94   | 88.86 | 91.81 | 88.70  | 94.20  | 82.84  | 90.31  | 85.31 | 88.38 |
> > > > |Sample-wise| Ours| 15.36   | 10.79  | 10.00  | 14.72 | 17.68 | 17.00   | 21.12 | 17.61 | 22.78  | 11.16  | 80.41  | 81.03  | 38.31 | **27.54** |
> > > > | |  |||||||||||||||
> > > > | Class-wise  | Next Class| 10.00   | 10.00  | 9.96   | 10.18 | 10.47 | 10.18   | 10.00 | 10.00 | 10.00  | 10.00  | 62.91  | 17.26  | 10.01 | 14.69 |
> > > > | Class-wise  | Random Natural Images | 10.00   | 10.00  | 11.41  | 10.00 | 10.01 | 11.71   | 10.00 | 10.01 | 10.00  | 10.00  | 58.38  | 10.05  | 10.00 | **13.97** |
> > > > |Class-wise  | Ours | 10.00   | 10.00  | 11.25  | 10.02 | 10.59 | 10.04   | 13.53 | 10.00 | 10.00  | 10.00  | 72.97  | 23.62  | 10.00 | 16.31 |

---

### Official Review · Reviewer_4Lh6 · 2023-10-29

**Soundness:** 2 fair
**Presentation:** 2 fair
**Contribution:** 2 fair
**Rating:** 5
**Confidence:** 3

**Summary:**

This paper focuses on methods for generating unlearnable samples. The paper takes the unique perspective of deep hiding and improves based on existing deep hiding methods, which consequently proposes new methods for generating unlearnable samples. Experiments demonstrate the effectiveness and robustness of the proposed method.

**Strengths:**

1. The paper is well written and has clear figures.

2. The paper proposed methods for generating unlearnable samples from such an interesting perspective as deep hiding.

3. The experiments demonstrate the high effectiveness and robustness of the method proposed in the paper.

**Weaknesses:**

1. The comparison experiments do not show the relationship between the scale of the perturbations generated by the proposed method and the comparison method, and the size of the generated perturbations cannot be fully controlled by $L_{hide}$ alone, because there are other losses included in the total loss.

2. An ablation experiment on the semantic image generation module is missing to demonstrate its advantages compared with randomly selected hidden images.

**Questions:**

1. There is a backward revealing process for the INN-based hiding model used in the paper, but what is the significance of the existence of this process?

2. The paper proposed a concentration loss (Eq. 7), and how are sample i and sample j selected in the specific implementation?


======================After rebuttal===================

The authors' response address most of my concerns. Thus I am willing to increase the rating score to 6.


======================Update after discussion===================

After discussion, I agree with reviewer Naq4. For data with larger resolution, while it contains more information that needs to be protected, there are also more features that can be used to hide critical content. However, the proposed method seems to be limited on more complex datasets (e.g., ImageNet-subset). Thus, I think the current work needs further improvements to meet the acceptance criteria.

---

> ### Author Response · Authors · 2023-11-20
>
> **Q1:** The comparison experiments do not show the relationship between the scale of the perturbations generated by the proposed method and the comparison method, and the size of the generated perturbations cannot be fully controlled by alone, because there are other losses included in the total loss.
>
> **Ans:** Thank you for your insightful comment. We evaluate the image quality (PSNR) between the generated unlearnable examples and clean image to analysize the perturbation scale. The PSNR results are: Ours (34.04), EM(34.17), REM(33.96), TAP(35.86) and LSP(37.73). This similarity in image quality indicates our method achieve comparable scale of the pertuibations compared to comparison methods.
> Despite we did not solely control the perturbation loss, this result further demonstrates that our perturbation loss incorporates well with other losses to restrict the perturbation scale (8/255) during joint optimization.
>
> **Q2:** An ablation experiment on the semantic image generation module is missing to demonstrate its advantages compared with randomly selected hidden images.
>
> **Ans:** Thank you for the comments. We conduct a comprehensive ablation study on the effectiveness of the semantic image generation module. We show the results in Table 4 and we have added it into the paper. In the class-wise setting, we find that the improvement of the generation module is marginal; however, in the sample-wise setting, the semantic generation module (SIG) can degrade the mean accuracy from 88% to 38%. It is further reduced to 27.5% with latent feature concentration (LFC). When we disentangle the text prompt clustering (TPC) and Stable Diffusion+ControlNet generation (SD+C), we find that SD+C contributes the most and TPC contributes around a 2% reduction. The ablation study shows that each component in our proposed method plays an important role in the generally robust unlearnable examples.
>
> **Table 4: Ablation studies on CIFAR10 for designed Latent Feature Concentration module (LFC), and Semantic Images Generation module (SIG), including Text Prompts Clustering (TPC) and Stable Diffusion model and ControlNet (SD+C).**
>
> | Setting | LFC | SIG: TPC | SIG: SD+C | Vanilla | Cutout | Cutmix | Mixup | MeanF | MedianF | BDR   | Gray  | GaussN | GaussF | JPEG10 | JPEG50 | AT    | Mean  |
> |:-------:|:---:|:--------:|:---------:|:-------:|:------:|:------:|:-----:|:-----:|:-------:|:-----:|:-----:|:------:|:------:|:------:|:------:|:-----:|:-----:|
> | Sample-wise|   ×  |       ×   |        ×   | 94.08   | 94.52  | 94.07  | 88.23 | 65.42 | 87.04   | 89.46 | 91.45 | 88.36  | 94.07  | 83.29  | 89.97  | 86.26 | 88.17 |
> |    Sample-wise     | ×    | ✓        | ✓         | 14.77   | 22.20  | 13.06  | 23.44 | 30.18 | 51.43   | 35.66 | 17.31 | 37.50  | 15.80  | 81.97  | 81.48  | 81.24 | 38.93 |
> |  Sample-wise       | ✓   |    ×       | ✓         | 10.00   | 16.53  | 20.81  | 17.14 | 18.51 | 21.73   | 24.98 | 13.85 | 22.07  | 10.59  | 80.09  | 82.90  | 46.54 | 29.67 |
> |    Sample-wise     | ✓   |    ×       |    ×   | 94.09   | 94.34  | 93.84  | 94.00 | 64.55 | 85.94   | 88.86 | 91.00 | 88.70  | 94.20  | 82.84  | 90.31  | 85.31 | 88.38 |
> |     Sample-wise    | ✓   | ✓  | ✓  | 15.36   | 10.79  | 10.00  | 14.72 | 17.68 | 17.00   | 21.12 | 17.61 | 22.78  | 11.16  | 80.41  | 81.03  | 38.31 | **27.54** |
> | Class-wise |  ×    |     ×      |    ×        | 12.16   | 11.02  | 11.43  | 10.81 | 10.14 | 13.28   | 10.10 | 12.12 | 10.00  | 10.00  | 78.21  | 15.32  | 12.87 | 16.73 |
> |    Class-wise     |  ×    | ✓   | ✓   | 10.00   | 10.00  | 10.06  | 10.82 | 13.26 | 25.76   | 15.83 | 14.20 | 10.03  | 10.00  | 76.26  | 20.28  | 14.78 | 18.56 |
> |    Class-wise     | ✓   |  ×     | ✓   | 10.00   | 10.00  | 10.00  | 12.32 | 9.72  | 10.00   | 10.00 | 10.59 | 10.00  | 10.00  | 71.44  | 30.27  | 10.00 | 16.49 |
> |    Class-wise     | ✓   |  ×  |    ×  | 10.00   | 10.00  | 11.41  | 10.00 | 10.01 | 11.71   | 10.00 | 10.01 | 10.00  | 10.00  | 58.38  | 10.05  | 10.00 | **13.97** |
> |    Class-wise     | ✓   | ✓ | ✓  | 10.00   | 10.00  | 11.25  | 10.02 | 10.59 | 10.04   | 13.53 | 10.00 | 10.00  | 10.00  | 72.97  | 23.62  | 10.00 | 16.31 |
>
> **Q3:** There is a backward revealing process for the INN-based hiding model used in the paper, but what is the significance of the existence of this process?
>
> **Ans:** The backward revealing process is necessary to ensure that the semantic image is successfully hidden into the clean images.
> In the case of without revealing process, the network tends to embed nothing for satisfying the invisibility.
> In practice, during inference when generating UE, we do not require the revealing process but only utilize the hiding process.
>
> **Q4:** The paper proposed a concentration loss (Eq. 7), and how are sample i and sample j selected in the specific implementation?
>
> **Ans:** Sample ij refers to two samples within the same minibatch that have the same label. This is because our LFC only operates on pairs of samples with matching labels.

---

> > ### Comment · Reviewer_4Lh6 · 2023-11-22
> > **Response to authors**
> >
> > Thanks for the authors' respnse. Although the authors provide some results and explanations, I still believe that solely controlling the perturbation loss is useful to comprehensively present the effects of the proposed method. I am sorry that the I will maintain the original rating score.

---

> ### Author Response · Authors · 2023-11-22
>
> Thank you for your prompt response and insightful speculation.
>
> To demonstrate the impact of solely training with perturbation loss (i.e., **$\mathcal{L}_{hide}$**), we conduct relevant experiments. **1.** The training plot (Figure A in the supp) shows that the perturbation loss remains at 0, which indicates there is no optimization in hiding semantic images. **2.** The perturbation maps generated by the hiding model trained on only **$\mathcal{L}_{hide}$** ( Figure B in the supp) show minimal information with no semantic patterns (i.e., black images). **3.** Our evaluation of unlearnable examples generated with only **$\mathcal{L}_{hide}$** reveals that the test accuracy is close to that of clean images (Table. K). This indicates that minimal information is hidden in clean images, leading to ineffective unlearnability.
>
> **Table. K: Evaluation of the different hiding models trained by solely controlling the hiding loss (**$\mathcal{L}_{\text{hide}}$**) and using our designed loss ($\mathcal{L}_{\text{total}}$). The test accuracy (\%) is evaluated on CIFAR-10 in the class-wise setting.**
> | Method | Vanilla | Cutout | Cutmix | Mixup | MeanF | MedianF | BDR | Gray | GaussN | GaussF | JPEG10 | JPEG50 | AT | Mean |
> | :---: | :---: | :---: | :---: | :---: | :---: | :---: | :---: | :---: | :---: | :---: | :---: | :---: | :---: | :---: |
> | Clean | 94.59 | 95.00 | 94.77 | 94.96 | 49.70 | 86.64 | 89.07 | 92.80 | 88.71 | 94.54 | 85.22 | 90.89 | 84.19 | 87.78 |
> | only $\mathcal{L}_{\text {hide }}$ | 94.71 | 95.05 | 94.68 | 95.4 | 38.88 | 87.39 | 89.12 | 93.1 | 88.55 | 87.39 | 85.04 | 91.04 | 88.59 | 86.84 |
> | ours | 10.00 | 10.00 | 11.25 | 10.02 | 10.59 | 10.04 | 13.53 | 10.00 | 10.00 | 10.00 | 72.97 | 23.62 | 10.00 | **16.31** |
>
> This phenomenon is because only controlling the perturbation loss (i.e., $\mathcal{L}_{hide}$) alone will lead to unstructured minimal perturbation. When solely optimizing the perturbation loss, the network tends to discard the information of the semantic image to meet the requirement of minimizing the difference between the clean image and generated UE rather than hide it. Thus, the revealing loss is important to enforce the network to hide the information of semantic image into the clean image by ensuring the revealing of the hidden information from the UE.
>
> Based on the experimental results and our analysis, we argue that the revealing process with reveal loss is necessary to hide semantic information to create UEs. We hope it can address your concerns.

---

> > ### Comment · Reviewer_4Lh6 · 2023-11-23
> > **Response to authors**
> >
> > Thanks for your quick response. Based on the results you provided, I recognize the validity of the proposed method. Thus, I am willing to increase the rating score.

---

### Official Review · Reviewer_Naq4 · 2023-10-29

**Soundness:** 2 fair
**Presentation:** 3 good
**Contribution:** 2 fair
**Rating:** 5
**Confidence:** 4

**Summary:**

The paper proposes a novel method to generate robust unlearnable examples by hiding semantic images within clean images using invertible neural networks (INNs). This introduces perturbations to mislead classifiers while leveraging semantic features that are robust to countermeasures. A Latent Feature Concentration (LFC) module regularizes the intra-class variance of perturbations. A Semantic Images Generation module creates hidden images with consistent semantics within a class to maximize inter-class separation. Experiments on CIFAR and ImageNet datasets demonstrate state-of-the-art unlearnability and robustness against data augmentations and preprocessing.

**Strengths:**

1. Novel deep hiding scheme to generate unlearnable examples by hiding semantic images using INNs.

2. Introduces LFC module to regularize intra-class variance of perturbations.

3. Introduces Semantic Image Generation module to maximize inter-class separation.

4. State-of-the-art results on CIFAR and ImageNet datasets against various countermeasures.

**Weaknesses:**

1. Additional Requirements of generating a large dataset of semantic images using paired text prompts and canny edge maps.

2. The sample-wise setting may leak information about hidden images. If some hackers know an image is protected in this way, they may find a countermeasure for the unlearnable examples based on the proposed hidden semantic generations.

3. A pre-trained ResNet-18 is used as the feature extractor, all text prompts are clustered using K-means with the semantic features from the CLIP model, and Stable Diffusion model and ControlNet to generate semantic images. It's hard to analyze the effectiveness of each component.

4. From the above, the connection between the hidden image and the unlearnable example is unclear. The hidden semantic image may not directly contribute to unlearnability.

5. The robustness and effectiveness may come from the pre-trained ResNet-18, CLIP, Stable Diffusion, or ControlNet. Therefore, the effectiveness and robustness may not come from the architecture and the idea; instead, these may only come from the extra information from the four pre-trained models.

6. Although an LFC (a pretrained ResNet18) is used to regularize intra-class variance, and the CLIP, Stable Diffusion, and ControlNet are used to maximize inter-class separation, the author should consider the alignment between these pretrained models.

7. Although Grad-CAM is used to visualize the attention of DNNs, the intra-class and inter-class relationship with the classification and semantic generation should be more fully exploited. Evaluating the inter-class and intra-class statistics directly could also substantiate the claims around controlled semantics, see questions.

**Questions:**

On page 4, the paper mentions the previous work lacks semantic high-level features and redundancy. However, I don’t know how redundancy is solved in this work.

There are some ablation experiments that remove each major component would indeed provide better insights into their individual contributions:

1. Using clean images from different classes as hidden semantic images, or using random natural images as hidden semantic images, rather than generated ones. As you noted, this removes the control over the consistency of semantics within a class. The drop in unlearnability can show the importance of controlled generation.

2. Removing the Latent Feature Concentration (LFC) module. This would demonstrate the impact of the proposed module in regularizing intra-class perturbations.

3. Removing the CLIP-based clustering of text prompts. Using random prompts for generation removes controlled inter-class differences.

4. Evaluating inter-class and intra-class separation quantitatively using metrics like mean intra-class distance and mean inter-class distance. This can formally validate the claims.

---

> ### Author Response · Authors · 2023-11-20
>
> **Q1:** Additional Requirements of generating a large dataset of semantic images using paired text prompts and canny edge maps.
>
>
> **Ans:** Thank you for your invaluable comments on the ‘semantic image generation’ module. As we mention in Section 3.2.3, we use the paired text prompt and canny edge maps to control the generated semantic content more precisely, which leads to better robustness of the UEs.
> We conduct ablation studies using random natural images, and next-class images, without using paired text prompts and canny edge maps.
> Table D demonstrates that without the ControlNet based generation, the performance is still competitive in the class-wise setting while dropping dramatically in the sample-wise setting.
> The competitive results in the class-wise setting prove the effectiveness of unlearnability brought from hiding semantic images, highlighting robust generalizability across various countermeasures.
> However, simply using next-class images or random natural images in the sample-wise setting fails to achieve unlearnability due to without controlling the similarity of hidden semantic images within intra-class data, which validates the effectiveness of generating a large dataset of semantic images using paired text prompts and canny edge maps.
>
> **Table D: Additional experimental results on CIFAR10 by using different hidden semantic images, including images from the next class, random natural images (CIFAR100), and our semantic image generation module.**
>
> | Setting         | Hidden Semantic Images       | Vanilla | Cutout | Cutmix | Mixup | MeanF | MedianF | BDR   | Gray  | GaussN | GaussF | JPEG10 | JPEG50 | AT    | Mean  |
> |:---------------:|:---------------------:|:-------:|:------:|:------:|:-----:|:-----:|:-------:|:-----:|:-----:|:------:|:------:|:------:|:------:|:-----:|:-----:|
> | Sample-wise | Next Class            | 80.19   | 75.79  | 74.79  | 71.26 | 60.57 | 82.25   | 72.89 | 67.46 | 72.38  | 78.66  | 82.01  | 87.68  | 82.53 | 76.04 |
> |      Sample-wise           | Random Natural Images | 94.09   | 94.34  | 93.84  | 94.17 | 64.55 | 85.94   | 88.86 | 91.81 | 88.70  | 94.20  | 82.84  | 90.31  | 85.31 | 88.38 |
> |      Sample-wise           | Ours                  | 15.36   | 10.79  | 10.00  | 14.72 | 17.68 | 17.00   | 21.12 | 17.61 | 22.78  | 11.16  | 80.41  | 81.03  | 38.31 | **27.54** |
> |                 |                       |         |        |        |       |       |         |       |       |        |        |        |        |       |       |
> | Class-wise  | Next Class            | 10.00   | 10.00  | 9.96   | 10.18 | 10.47 | 10.18   | 10.00 | 10.00 | 10.00  | 10.00  | 62.91  | 17.26  | 10.01 | 14.69 |
> |         Class-wise        | Random Natural Images | 10.00   | 10.00  | 11.41  | 10.00 | 10.01 | 11.71   | 10.00 | 10.01 | 10.00  | 10.00  | 58.38  | 10.05  | 10.00 | **13.97** |
> |         Class-wise        | Ours                  | 10.00   | 10.00  | 11.25  | 10.02 | 10.59 | 10.04   | 13.53 | 10.00 | 10.00  | 10.00  | 72.97  | 23.62  | 10.00 | 16.31 |
>
> **Q2:** The sample-wise setting may leak information about hidden images. If some hackers know an image is protected in this way, they may find a countermeasure for the unlearnable examples based on the proposed hidden semantic generations.
>
> **Ans:** Thank you for your insights. We would like to clarify that we reduce the exposure risk of the sample-wise setting from 2 aspects:
>
> 1. We hide different semantic images into the images in each class. Though the semantic images share similar semantics, they are totally different in textures and colors.
> 2. Our deep hiding model will generate content-dependent perturbations. When we hide the semantic image into the original image, the perturbation will embed into the original content adaptively, leading to more pixel-wise changes.
>
> In data protection and adversarial learning research, we will always face challenges from hackers who wish to mitigate the effects of the perturbations. We are consistently improving the robustness of the defensive perturbations, so the hackers will need more effort to remove them. It is the initiative of our work as well, and we believe the research community is working towards more generally robust unlearnable examples endlessly.

---

> ### Author Response · Authors · 2023-11-20
>
> **Q3:** A pre-trained ResNet-18 is used as the feature extractor, all text prompts are clustered using K-means with the semantic features from the CLIP model, and Stable Diffusion model and ControlNet to generate semantic images. It's hard to analyze the effectiveness of each component.
>
> **Ans:** Thank you for the comments. We conduct a comprehensive ablation study on the effectiveness of each component. We show the results in Table 4, and we have added it into the paper. In the class-wise setting, we find that the improvement of the generation module is marginal; however, in the sample-wise setting, the semantic generation module (SIG) can degrade the mean accuracy from 88% to 38%. It is further reduced to 27.5% with latent feature concentration (LFC). When we disentangle the text prompt clustering (TPC) and Stable Diffusion+ControlNet generation (SD+C), we find that SD+C contributes the most and TPC causes around a 2% reduction. The ablation study shows that each component in our proposed method plays an important role in the generally robust unlearnable examples.
>
> **Table 4: Ablation studies on CIFAR10 for designed Latent Feature Concentration module (LFC), and Semantic Images Generation module (SIG), including Text Prompts Clustering (TPC) and Stable Diffusion model and ControlNet (SD+C).**
>
> | Setting | LFC | SIG: TPC | SIG: SD+C | Vanilla | Cutout | Cutmix | Mixup | MeanF | MedianF | BDR   | Gray  | GaussN | GaussF | JPEG10 | JPEG50 | AT    | Mean  |
> |:-------:|:---:|:--------:|:---------:|:-------:|:------:|:------:|:-----:|:-----:|:-------:|:-----:|:-----:|:------:|:------:|:------:|:------:|:-----:|:-----:|
> | Sample-wise|   ×  |       ×   |        ×   | 94.08   | 94.52  | 94.07  | 88.23 | 65.42 | 87.04   | 89.46 | 91.45 | 88.36  | 94.07  | 83.29  | 89.97  | 86.26 | 88.17 |
> |    Sample-wise     | ×    | ✓        | ✓         | 14.77   | 22.20  | 13.06  | 23.44 | 30.18 | 51.43   | 35.66 | 17.31 | 37.50  | 15.80  | 81.97  | 81.48  | 81.24 | 38.93 |
> |  Sample-wise       | ✓   |    ×       | ✓         | 10.00   | 16.53  | 20.81  | 17.14 | 18.51 | 21.73   | 24.98 | 13.85 | 22.07  | 10.59  | 80.09  | 82.90  | 46.54 | 29.67 |
> |    Sample-wise     | ✓   |    ×       |    ×        | 94.09   | 94.34  | 93.84  | 94.00 | 64.55 | 85.94   | 88.86 | 91.00 | 88.70  | 94.20  | 82.84  | 90.31  | 85.31 | 88.38 |
> |     Sample-wise    | ✓   | ✓        | ✓         | 15.36   | 10.79  | 10.00  | 14.72 | 17.68 | 17.00   | 21.12 | 17.61 | 22.78  | 11.16  | 80.41  | 81.03  | 38.31 | **27.54** |
> | Class-wise |  ×    |     ×      |    ×        | 12.16   | 11.02  | 11.43  | 10.81 | 10.14 | 13.28   | 10.10 | 12.12 | 10.00  | 10.00  | 78.21  | 15.32  | 12.87 | 16.73 |
> |    Class-wise     |  ×    | ✓        | ✓         | 10.00   | 10.00  | 10.06  | 10.82 | 13.26 | 25.76   | 15.83 | 14.20 | 10.03  | 10.00  | 76.26  | 20.28  | 14.78 | 18.56 |
> |    Class-wise     | ✓   |      ×     | ✓         | 10.00   | 10.00  | 10.00  | 12.32 | 9.72  | 10.00   | 10.00 | 10.59 | 10.00  | 10.00  | 71.44  | 30.27  | 10.00 | 16.49 |
> |    Class-wise     | ✓   |     ×      |    ×        | 10.00   | 10.00  | 11.41  | 10.00 | 10.01 | 11.71   | 10.00 | 10.01 | 10.00  | 10.00  | 58.38  | 10.05  | 10.00 | **13.97** |
> |    Class-wise     | ✓   | ✓        | ✓         | 10.00   | 10.00  | 11.25  | 10.02 | 10.59 | 10.04   | 13.53 | 10.00 | 10.00  | 10.00  | 72.97  | 23.62  | 10.00 | 16.31 |
>
> **Q4:** From the above, the connection between the hidden image and the unlearnable example is unclear. The hidden semantic image may not directly contribute to unlearnability.
>
> **Ans:** We agree that it is difficult to give a certain answer about the source of unlearnability, since the image generation and hiding network are still black boxes. We want to reclaim our hypothesis in Section 3.3 that deep hiding semantic images can create shortcuts, leading to unlearnability. Furthermore, based on the ablation results in Table D(**Q1**) and Table 4(**Q3**), we show the hidden semantic image contributes to the unlearnability in the class-wise setting; In the sample-wise setting, the unlearnability mainly comes from the controlled semantic image generation. These results highlight the effectiveness of the hidden semantic image.
>
> **Q5:** The robustness and effectiveness may come from the pre-trained ResNet-18, CLIP, Stable Diffusion, or ControlNet. Therefore, the effectiveness and robustness may not come from the architecture and the idea; instead, these may only come from the extra information from the four pre-trained models.
>
> **Ans:** Similar to the previous responses, we would like to clarify that the effectiveness of the unlearnable examples comes from the semantic hiding images instead of the pre-trained models, based on the analysis and experimental results.

---

> ### Author Response · Authors · 2023-11-20
>
> **Q6:** Although an LFC (a pretrained ResNet18) is used to regularize intra-class variance, and the CLIP, Stable Diffusion, and ControlNet are used to maximize inter-class separation, the author should consider the alignment between these pretrained models.
>
> **Ans:** We've conducted an ablation study by removing the LFC (Table 4(Q3)) and found that there is a slight decrease on the experimental results.
> The reason is that these components serve as auxiliary elements to control the semantic image. Therefore, alignment is not an urgent requirement and does not significantly affect the results.
>
> **Q7:** Although Grad-CAM is used to visualize the attention of DNNs, the intra-class and inter-class relationship with the classification and semantic generation should be more fully exploited. Evaluating the inter-class and intra-class statistics directly could also substantiate the claims around controlled semantics, see questions.
>
> **Ans:** We investigate the intra-/inter-class distance in latent features using a trained (for unlearnable examples) or pre-trained (for clean images) ResNet18, shown in Table J. We observe that our unlearnable examples, in both class-wise and sample-wise settings, exhibit significantly reduced intra-class distances, as evidenced by a higher cosine similarity approaching 1.0000. Furthermore, by compacting the semantics within each class, we also achieve an increased inter-class distance. These outcomes suggest that our method successfully generates unlearnable examples characterized by minimal intra-class distances and maximized inter-class distances, thereby enhancing unlearnability.
>
> The calculation detail of intra-/inter-class distance is shown below.
> We consider the output of the last CNN layer as the latent feature for each sample and average these to determine the mean latent feature per class. For intra-class distance, we compute and report the mean cosine similarity between each sample's latent feature and its class mean. For inter-class distance, we calculate the mean cosine similarity between each class's mean latent feature and the overall dataset mean.
>
> **Table J: Evaluation of Inter-Class and Intra-Class Distances.**
>
> | Data         | Intra-class | Inter-class |
> |:------------:|:-----------:|:-----------:|
> | Clean images | 0.8457      | 0.8742      |
> | Ours(S)      | 0.9702      | 0.9030      |
> | Ours(C)      | 0.9882      | 0.8815      |
>
> **Q8:** On page 4, the paper mentions the previous work lacks semantic high-level features and redundancy. However, I don’t know how redundancy is solved in this work.
>
> **Ans:** In our paper, we solve the redundancy issue by using the image-hiding framework, which can inherently and adaptively hide different amounts of information in each pixel.
> In this way, we can hide semantic images instead of noise-based perturbations in clean images. The redundancy found in semantic images – such as the repeated shapes and colors in a forest scene – allows for a certain degree of data loss or alteration without significantly impacting the image's overall structure or meaning. As a result, the high redundancy of the hidden semantic information contributes to the high tolerance of unlearnability when under image processing.
>
>
> **Q9:** Ablation studies:
> There are some ablation experiments that remove each major component would indeed provide better insights into their individual contributions:
> 1. Using clean images from different classes as hidden semantic images, or using random natural images as hidden semantic images, rather than generated ones. As you noted, this removes the control over the consistency of semantics within a class. The drop in unlearnability can show the importance of controlled generation.
> 2. Removing the Latent Feature Concentration (LFC) module. This would demonstrate the impact of the proposed module in regularizing intra-class perturbations.
> 3. Removing the CLIP-based clustering of text prompts. Using random prompts for generation removes controlled inter-class differences.
> 4. Evaluating inter-class and intra-class separation quantitatively using metrics like mean intra-class distance and mean inter-class distance. This can formally validate the claims.
>
> **Ans:** Thank you so much for the detailed suggestions on ablation studies. The ablations significantly improve our papers. We have conducted the following ablations and show the results in the respective tables:
> - Ablation on each component in Semantic Generation Module. The results are shown in Table D and Table 4.
> - Ablation on LFC. The results are shown in the modified Table 4.
>
> We hope ablation studies can address your concerns and provide better insights into the components.

---

### Official Review · Reviewer_p8VW · 2023-10-30

**Soundness:** 2 fair
**Presentation:** 3 good
**Contribution:** 2 fair
**Rating:** 6
**Confidence:** 5

**Summary:**

The paper presents an approach to generate unlearnable examples by hiding the semantic images in the natural images. To hide the semantic images, DWT along with INN has been used. The loss functions are combined to ensure a frequency image is close to the natural image and only high-frequency features are disturbed. The experiments are performed on multiple datasets and the resiliency of the proposed approach is also demonstrated.

**Strengths:**

The paper presents straightforward unlearnable examples generation algorithms.
The generated examples are robust to several defense strategies.

**Weaknesses:**

* The paper can bring the hiding process into the main text in comparison to adding it to the supplementary file.
* The experimental section of the paper is weak. The ablation studies with several parameters used in the proposed algorithm are missing. For example: ablation concerning the role of individual loss items, concerning values of $w_i$.
* The authors have mentioned that $100$ semantic images are generated for image hiding. Are these all $100$ images used and how?
* Why the existing algorithms are re-implemented? Are they implemented and fine-tuned using the parameters provided in the papers? can't we use the protocols of existing work for direct comparisons?
* Any reason why *JPEG10* is yielding significantly higher robustness across each unlearnable example? Any explainable reason for this and have the authors tried with a lower **JEPG** value?
* have the authors studied the transferability of the proposed examples in the presence of defenses?
* It is surprising to see that even at 80\%, the test accuracy is significantly high (84.40\%), which suddenly drops to 10.00\% in presence of 100\% examples. Am I correct? If yes, what is the reason?
* What about transferability in terms of limited samples used to learn unlearnable examples?
* The paper also needs to discuss existing contemporary image-hiding works effective in generating adversarial examples and how they are different from this work. Why they can not be used as compared to the proposed DH?

[1] Din SU, Akhtar N, Younis S, Shafait F, Mansoor A, Shafique M. Steganographic universal adversarial perturbations. Pattern Recognition Letters. 2020 Jul 1;135:146-52.

[2] A. Agarwal, N. Ratha, M. Vatsa and R. Singh, "Crafting Adversarial Perturbations via Transformed Image Component Swapping," in IEEE Transactions on Image Processing, vol. 31, pp. 7338-7349, 2022, doi: 10.1109/TIP.2022.3204206.

**Questions:**

Please check the weakness section.

---------------------------------- Post Rebuttal ----------------

The responses posted addressed my concerns.

---

> ### Author Response · Authors · 2023-11-20
>
> **Q1:** The paper can bring the hiding process into the main text in comparison to adding it to the supplementary file.
>
>
> **Ans:**  Thank you for your suggestion. We've added the specific hiding process to the main text, making it easier for readers to understand our method.
>
>
> **Q2:** The experimental section of the paper is weak. The ablation studies with several parameters used in the proposed algorithm are missing. For example: ablation concerning the role of individual loss items, concerning values of wi.
>
>
> **Ans:** Thank you for your comments on the experimental sections. We have added several ablation results based on your suggestions:
> - Ablation on weights of individual loss wi: we conduct a grid search on the 3 loss weights separately. We search from 0.01 to 10 for w1 and w2, and search from 1e-5 to 1e-2 for w3, since loss associated with w3 is generally 1000 times larger. The results in Table A&B&C show that when we set weights to 1, 1, 0.0001, the mean accuracy against countermeasures archives the best performance. Hence, we adopt these hyperparameters in all our experiments.
> - Ablation on generation module: we further conduct ablation study on the impact of the semantic image generation module (Table 4 and Table D). We show that each module plays significant role to further improve the robustness of the deep hiding unlearnable examples.
>
>
> We hope these amendments address your concerns and further strengthen our paper.
>
>
> **Table A: The experimental results of different settings on parameters of $\omega_1$.**
> | Setting     | $\omega_1$ | Vanilla | Cutout | Cutmix | Mixup | MeanF | MedianF | BDR   | Gray  | GaussN | GaussF | JPEG10 | JPEG50 | AT    | Mean      |
> |:-----------:|:----------:|:-------:|:------:|:------:|:-----:|:-----:|:-------:|:-----:|:-----:|:------:|:------:|:------:|:------:|:-----:|:---------:|
> | Sample-wise | 10         | 16.00   | 16.35  | 13.69  | 17.28 | 30.86 | 64.48   | 23.40 | 18.34 | 24.11  | 13.70  | 82.03  | 70.60  | 48.92 | 33.83     |
> |  Sample-wise  (Ours)         | 1    | 15.36   | 10.79  | 10.00  | 14.72 | 17.68 | 17.00   | 21.12 | 17.61 | 22.78  | 11.16  | 80.41  | 81.03  | 38.31 | **27.54** |
> |      Sample-wise       | $10^{-1}$  | 13.54   | 17.32  | 14.11  | 17.24 | 29.90 | 37.61   | 28.54 | 19.15 | 25.25  | 16.23  | 81.48  | 63.96  | 58.30 | 32.51     |
> |    Sample-wise         | $10^{-2}$  | 11.31   | 12.48  | 10.01  | 11.12 | 17.11 | 23.48   | 17.46 | 13.89 | 22.41  | 10.05  | 80.04  | 84.38  | 45.20 | 27.61     |
> | Class-wise  | 10         | 11.60   | 10.10  | 12.52  | 10.83 | 11.92 | 21.24   | 17.48 | 11.45 | 12.42  | 10.00  | 76.42  | 20.26  | 17.56 | 18.75     |
> |    Class-wise  (Ours)     | 1    | 10.00   | 10.00  | 11.25  | 10.02 | 10.59 | 10.04   | 13.53 | 10.00 | 10.00  | 10.00  | 72.97  | 23.62  | 10.00 | **16.31** |
> |      Class-wise     | $10^{-1}$  | 10.00   | 10.00  | 9.99   | 10.00 | 10.11 | 10.09   | 12.44 | 10.00 | 10.00  | 10.00  | 72.36  | 28.89  | 10.05 | 16.46     |
> |      Class-wise      | $10^{-2}$  | 10.00   | 10.00  | 10.02  | 10.00 | 10.09 | 10.00   | 16.34 | 10.00 | 10.00  | 10.00  | 72.45  | 58.50  | 10.00 | 19.03     |
>
> **Table B: The experimental results of different settings on parameters of $\omega_2$.**
> | Setting | $\omega_2$ | Vanilla | Cutout | Cutmix | Mixup | MeanF | MedianF | BDR   | Gray  | GaussN | GaussF | JPEG10 | JPEG50 | AT    | Mean      |
> |:-------:|:----------:|:-------:|:------:|:------:|:-----:|:-----:|:-------:|:-----:|:-----:|:------:|:------:|:------:|:------:|:-----:|:---------:|
> | Sample-wise  | 10         | 15.12   | 10.35  | 10.34  | 13.58 | 17.94 | 23.41   | 20.41 | 16.00 | 24.53  | 10.01  | 81.89  | 80.81  | 39.88 | 28.02     |
> |    Sample-wise (Ours)     | 1    | 15.36   | 10.79  | 10.00  | 14.72 | 17.68 | 17.00   | 21.12 | 17.61 | 22.78  | 11.16  | 80.41  | 81.03  | 38.31 | **27.54** |
> |    Sample-wise      | $10^{-1}$  | 16.99   | 16.81  | 11.01  | 12.22 | 25.72 | 35.44   | 26.33 | 20.33 | 27.34  | 16.66  | 82.40  | 81.74  | 69.43 | 34.03     |
> |     Sample-wise     | $10^{-2}$  | 10.57   | 15.16  | 21.60  | 14.12 | 19.67 | 35.44   | 22.87 | 18.17 | 29.63  | 10.82  | 82.99  | 87.50  | 76.03 | 34.20     |
> | Class-wise   | 10         | 10.00   | 10.00  | 11.22  | 11.94 | 10.09 | 10.10   | 13.27 | 10.00 | 10.00  | 10.08  | 77.23  | 22.81  | 10.12 | 16.68     |
> |  Class-wise  (Ours) | 1    | 10.00   | 10.00  | 11.25  | 10.02 | 10.59 | 10.04   | 13.53 | 10.00 | 10.00  | 10.00  | 72.97  | 23.62  | 10.00 | **16.31** |
> |   Class-wise      | $10^{-1}$  | 10.04   | 10.00  | 11.64  | 13.81 | 15.49 | 10.43   | 17.09 | 10.16 | 10.94  | 10.05  | 77.79  | 25.29  | 18.35 | 19.25     |
> |     Class-wise    | $10^{-2}$  | 10.00   | 10.00  | 10.03  | 10.28 | 10.41 | 10.36   | 15.29 | 10.87 | 10.00  | 10.00  | 79.46  | 29.51  | 10.00 | 17.40     |

---

> > ### Author Response · Authors · 2023-11-20
> >
> > **Table C: The experimental results of different settings on parameters of $\omega_3$.**
> > | Setting     | $\omega_3$       | Vanilla | Cutout | Cutmix | Mixup | MeanF | MedianF | BDR   | Gray  | GaussN | GaussF | JPEG10 | JPEG50 | AT    | Mean      |
> > |:-----------:|:----------------:|:-------:|:------:|:------:|:-----:|:-----:|:-------:|:-----:|:-----:|:------:|:------:|:------:|:------:|:-----:|:---------:|
> > | Sample-wise | $10^{-2}$        | 14.09   | 13.71  | 10.06  | 11.50 | 17.65 | 28.47   | 22.44 | 17.63 | 22.67  | 15.79  | 82.04  | 76.56  | 39.96 | 28.66     |
> > |     Sample-wise         | $10^{-3}$        | 29.41   | 21.50  | 17.33  | 31.99 | 71.78 | 71.78   | 57.44 | 30.15 | 48.16  | 22.85  | 82.41  | 84.34  | 84.63 | 50.29     |
> > |     Sample-wise  (Ours)   | $10^{-4}$  | 15.36   | 10.79  | 10.00  | 14.72 | 17.68 | 17.00   | 21.12 | 17.61 | 22.78  | 11.16  | 80.41  | 81.03  | 38.31 | **27.54** |
> > |      Sample-wise        | $10^{-5}$        | 11.10   | 14.35  | 13.08  | 19.27 | 23.38 | 40.61   | 47.25 | 28.73 | 53.54  | 10.62  | 83.18  | 89.32  | 83.69 | 39.86     |
> > | Class-wise  | $10^{-2}$        | 10.00   | 10.00  | 10.00  | 11.84 | 10.24 | 10.10   | 16.45 | 10.03 | 11.29  | 10.00  | 74.47  | 21.16  | 15.58 | 17.01     |
> > |      Class-wise        | $10^{-3}$        | 10.00   | 10.01  | 10.11  | 10.65 | 12.13 | 19.81   | 15.06 | 15.69 | 10.21  | 10.00  | 77.98  | 21.37  | 10.77 | 17.98     |
> > |     Class-wise  (Ours)   | $10^{-4}$  | 10.00   | 10.00  | 11.25  | 10.02 | 10.59 | 10.04   | 13.53 | 10.00 | 10.00  | 10.00  | 72.97  | 23.62  | 10.00 | **16.31** |
> > |       Class-wise       | $10^{-5}$        | 10.00   | 10.00  | 10.03  | 9.99  | 10.01 | 10.00   | 14.52 | 10.05 | 10.16  | 10.00  | 80.09  | 73.48  | 10.54 | 20.68     |
> >
> > **Table 4: Ablation studies on CIFAR10 for designed Latent Feature Concentration module (LFC) and Semantic Images Generation module (SIG, including Text Prompts Clustering (TPC) and Stable Diffusion model and ControlNet (SD+C)).**
> > | Setting | LFC | SIG: TPC | SIG: SD+C | Vanilla | Cutout | Cutmix | Mixup | MeanF | MedianF | BDR   | Gray  | GaussN | GaussF | JPEG10 | JPEG50 | AT    | Mean  |
> > |:-------:|:---:|:--------:|:---------:|:-------:|:------:|:------:|:-----:|:-----:|:-------:|:-----:|:-----:|:------:|:------:|:------:|:------:|:-----:|:-----:|
> > | Sample-wise|   ×  |       ×   |        ×   | 94.08   | 94.52  | 94.07  | 88.23 | 65.42 | 87.04   | 89.46 | 91.45 | 88.36  | 94.07  | 83.29  | 89.97  | 86.26 | 88.17 |
> > |    Sample-wise     | ×    | ✓        | ✓         | 14.77   | 22.20  | 13.06  | 23.44 | 30.18 | 51.43   | 35.66 | 17.31 | 37.50  | 15.80  | 81.97  | 81.48  | 81.24 | 38.93 |
> > |  Sample-wise       | ✓   |    ×       | ✓         | 10.00   | 16.53  | 20.81  | 17.14 | 18.51 | 21.73   | 24.98 | 13.85 | 22.07  | 10.59  | 80.09  | 82.90  | 46.54 | 29.67 |
> > |    Sample-wise     | ✓   |    ×       |    ×        | 94.09   | 94.34  | 93.84  | 94.00 | 64.55 | 85.94   | 88.86 | 91.00 | 88.70  | 94.20  | 82.84  | 90.31  | 85.31 | 88.38 |
> > |     Sample-wise (Ours)    | ✓   | ✓        | ✓         | 15.36   | 10.79  | 10.00  | 14.72 | 17.68 | 17.00   | 21.12 | 17.61 | 22.78  | 11.16  | 80.41  | 81.03  | 38.31 | **27.54** |
> > | Class-wise |  ×    |     ×      |    ×        | 12.16   | 11.02  | 11.43  | 10.81 | 10.14 | 13.28   | 10.10 | 12.12 | 10.00  | 10.00  | 78.21  | 15.32  | 12.87 | 16.73 |
> > |    Class-wise     |  ×    | ✓        | ✓         | 10.00   | 10.00  | 10.06  | 10.82 | 13.26 | 25.76   | 15.83 | 14.20 | 10.03  | 10.00  | 76.26  | 20.28  | 14.78 | 18.56 |
> > |    Class-wise     | ✓   |      ×     | ✓         | 10.00   | 10.00  | 10.00  | 12.32 | 9.72  | 10.00   | 10.00 | 10.59 | 10.00  | 10.00  | 71.44  | 30.27  | 10.00 | 16.49 |
> > |    Class-wise     | ✓   |     ×      |    ×        | 10.00   | 10.00  | 11.41  | 10.00 | 10.01 | 11.71   | 10.00 | 10.01 | 10.00  | 10.00  | 58.38  | 10.05  | 10.00 | **13.97** |
> > |    Class-wise (Ours)    | ✓   | ✓        | ✓         | 10.00   | 10.00  | 11.25  | 10.02 | 10.59 | 10.04   | 13.53 | 10.00 | 10.00  | 10.00  | 72.97  | 23.62  | 10.00 | 16.31 |

---

> ### Author Response · Authors · 2023-11-20
>
> **Table D: Additional experimental results on CIFAR10 by using different hidden semantic images, including images from the next class, random natural images (CIFAR100), and our semantic image generation module.**
> | Setting         | Hidden Semantic Images       | Vanilla | Cutout | Cutmix | Mixup | MeanF | MedianF | BDR   | Gray  | GaussN | GaussF | JPEG10 | JPEG50 | AT    | Mean  |
> |:---------------:|:---------------------:|:-------:|:------:|:------:|:-----:|:-----:|:-------:|:-----:|:-----:|:------:|:------:|:------:|:------:|:-----:|:-----:|
> | Sample-wise | Next Class            | 80.19   | 75.79  | 74.79  | 71.26 | 60.57 | 82.25   | 72.89 | 67.46 | 72.38  | 78.66  | 82.01  | 87.68  | 82.53 | 76.04 |
> |      Sample-wise           | Random Natural Images | 94.09   | 94.34  | 93.84  | 94.17 | 64.55 | 85.94   | 88.86 | 91.81 | 88.70  | 94.20  | 82.84  | 90.31  | 85.31 | 88.38 |
> |      Sample-wise           | Ours                  | 15.36   | 10.79  | 10.00  | 14.72 | 17.68 | 17.00   | 21.12 | 17.61 | 22.78  | 11.16  | 80.41  | 81.03  | 38.31 | **27.54**|
> | Class-wise  | Next Class            | 10.00   | 10.00  | 9.96   | 10.18 | 10.47 | 10.18   | 10.00 | 10.00 | 10.00  | 10.00  | 62.91  | 17.26  | 10.01 | 14.69 |
> |         Class-wise        | Random Natural Images | 10.00   | 10.00  | 11.41  | 10.00 | 10.01 | 11.71   | 10.00 | 10.01 | 10.00  | 10.00  | 58.38  | 10.05  | 10.00 | **13.97** |
> |         Class-wise        | Ours                  | 10.00   | 10.00  | 11.25  | 10.02 | 10.59 | 10.04   | 13.53 | 10.00 | 10.00  | 10.00  | 72.97  | 23.62  | 10.00 | 16.31 |
>
> **Q3:** The authors have mentioned that 100 semantic images are generated for image hiding. Are these all 100 images used and how?
>
> **Ans:** We generate 100 images for each class, resulting in a total of 1k, 10k, and 10k semantic images for CIFAR10, CIFAR100, and ImageNet-100, respectively. Thus, these semantic images have the same class number as the target dataset.
>
> During the hiding process, we differ in the image selection for two types of hiding settings:
>
> - In the class-wise setting, we randomly select only one semantic image for all the clean images in the same class. In other words, each image in a class shares the same selected semantic images, while images in different classes are assigned with different semantic images.
>
> - In the sample-wise setting, for each class, we generate 100 semantic images with the same text prompt and canny edge map, to balance the diversity and generation efficiency of the semantic images. For each image in the same class, we randomly pick one semantic image from the 100-image pool and hide it into the clean image.
>
> **Q4:** Why the existing algorithms are re-implemented? Are they implemented and fine-tuned using the parameters provided in the papers? can't we use the protocols of existing work for direct comparisons?
>
>
> **Ans:** Thanks for your question regarding the re-implementation of existing algorithms.
> We would like to clarify that we follow the same protocols of existing works based on their papers and the public code repositories. Besides, we implement and fine-tune the parameters provided in the papers. The ‘re-implementation’ indicates that we integrate all the existing methods into a unified code framework.
>
> The primary reason for ‘re-implementing’ these algorithms was to ensure consistency and fairness in our comparisons. To evaluate the general robustness of the UEs, our study involved 13 different countermeasures. Since the existing papers did not provide all the necessary results for these specific conditions, we opted to implement the existing methods in a unified framework.
>
> **Q5:** Any reason why JPEG10 is yielding significantly higher robustness across each unlearnable example? Any explainable reason for this and have the authors tried with a lower JEPG value?
>
> **Ans:** We show examples of images compressed by JPEG(quality factor=10) in the supplementary. The examples show that the compression level of JPEG10 can result in significant distortion to the images, leading to a decrease in the performance of UE.
> In response to your suggestion, we experimented with lower JPEG values (Table I). We found that lower JPEG values result in even lower test accuracy, which confirms that stronger compression not only damages the unlearnable example perturbations but also distorts the original image features significantly.
>
> **Table I: Test accuracy (\%) of model train on unlearnable examples from CIFAR10 against JPEG compression.**
> | JPEG Quality Factor |2|     4     | 6     | 8     | 10    | 50    |
> |:-------------------:|:-----:|:-----:|:-----:|:-----:|:-----:|:-----:|
> | Ours(S)         | 68.01 | 74.35 | 77.39 | 78.56 | 80.41 | 81.03 |
> | Ours(C)      | 64.53 | 70.18 | 72.56 | 72.72 | 72.97 | 23.62 |

---

> > ### Author Response · Authors · 2023-11-20
> >
> > **Q6:** Have the authors studied the transferability of the proposed examples in the presence of defenses?
> >
> > **Ans:** To ensure a thorough validation, we tested our UEs on CIFAR10 across multiple architectures, which provided us with a broad view of their transferability.
> > The detailed results have also been included in the supplementary file for reference. This additional data supports our claim that the proposed deep hiding UEs maintain their efficacy against different countermeasures across architectures.
> >
> > **Table E: Test accuracy (\%) of model train on unlearnable examples from CIFAR-10 with five architectures, including ResNet-18 (R18), ResNet-50 (R50), VGG-19 (V19), and DenseNet-121 (D121), and Vision Transformer (ViT), against data augmentations, data preprocessing, and adversarial training.**
> >
> > | Settings    | Model | Vanilla | Cutout | Cutmix | Mixup | MeanF | MedianF | BDR   | Gray  | GaussN | GaussF | JPEG10 | JPEG50 | AT    | Mean  |
> > |:-----------:|:-----:|:-------:|:------:|:------:|:-----:|:-----:|:-------:|:-----:|:-----:|:------:|:------:|:------:|:------:|:-----:|:-----:|
> > | Ours(S) | R18   | 15.36   | 10.79  | 10.00  | 14.72 | 17.68 | 17.00   | 21.12 | 17.61 | 22.78  | 11.16  | 80.41  | 81.03  | 38.31 | 27.54 |
> > |    Ours(S)         | R50   | 13.32   | 12.42  | 12.99  | 11.60 | 18.70 | 22.70   | 23.77 | 12.62 | 17.24  | 16.23  | 80.70  | 78.30  | 37.11 | 27.52 |
> > |     Ours(S)        | V19   | 10.45   | 17.25  | 14.37  | 17.87 | 23.62 | 32.27   | 22.77 | 17.72 | 22.28  | 14.89  | 80.61  | 80.64  | 52.55 | 31.33 |
> > |      Ours(S)       | D121  | 18.88   | 21.16  | 12.52  | 18.81 | 53.41 | 53.10   | 28.10 | 18.06 | 12.54  | 18.22  | 77.93  | 76.46  | 70.58 | 36.91 |
> > |      Ours(S)       | ViT   | 15.80   | 20.64  | 21.93  | 10.67 | 54.05 | 54.57   | 25.40 | 15.20 | 22.45  | 21.38  | 65.38  | 64.91  | 49.50 | 33.99 |
> > | Ours(C)  | R18   | 10.00   | 10.00  | 11.25  | 10.02 | 10.59 | 10.04   | 13.53 | 10.00 | 10.00  | 10.00  | 72.97  | 23.62  | 10.00 | 16.31 |
> > |     Ours(C)        | R50   | 10.00   | 10.00  | 11.02  | 10.04 | 10.06 | 10.45   | 17.31 | 10.00 | 10.00  | 10.00  | 74.43  | 24.15  | 11.09 | 16.81 |
> > |   Ours(C)            | V19   | 10.58   | 10.17  | 10.00  | 17.46 | 10.78 | 12.86   | 15.79 | 10.30 | 10.03  | 10.03  | 71.77  | 24.84  | 13.44 | 17.54 |
> > |      Ours(C)         | D121  | 10.00   | 10.00  | 10.60  | 10.23 | 10.49 | 10.01   | 11.63 | 10.00 | 10.00  | 10.53  | 72.85  | 22.06  | 10.00 | 16.03 |
> > |    Ours(C)           | ViT   | 10.00   | 10.01  | 10.02  | 10.67 | 11.23 | 22.34   | 12.35 | 10.12 | 10.00  | 10.00  | 61.92  | 30.69  | 18.79 | 17.55 |
> >
> > **Q7:** It is surprising to see that even at 80%, the test accuracy is significantly high (84.40%), which suddenly drops to 10.00% in presence of 100% examples. Am I correct? If yes, what is the reason?
> >
> > **Ans:** You are right. We hypothesize that this is because 20% of clean data provides useful features to guide the classifier, thereby affecting the shortcut learning towards hidden semantic images. This phenomenon aligns with the results of previously published papers on unlearnable examples [A][B][C]. In existing methods, a significant drop in performance has been observed for a subset of data subjected to perturbations.
> > ```
> > [A] Hanxun Huang, Xingjun Ma, Sarah Monazam Erfani, James Bailey, and Yisen Wang. Unlearnable examples: Making personal data unexploitable. ICLR, 2021.
> > [B] Shaopeng Fu, Fengxiang He, Yang Liu, Li Shen, and Dacheng Tao. Robust unlearnable examples: Protecting data privacy against adversarial learning. ICLR, 2022.
> > [C] Sadasivan, Vinu Sankar, Mahdi Soltanolkotabi, and Soheil Feizi. Cuda: Convolution-based unlearnable datasets. CVPR, 2023.
> > ```
> > **Table G: Test accuracy (\%) of CIFAR-10 on the models trained by
> > the clean data mixed with different percentages of unlearnable examples.**
> > | Percentage  | 20%    | 40%    | 60%    | 80%    | 85%    | 90%    | 92%    | 94%    | 96%    | 98%    | 100%   |
> > |:-----------:|:-----:|:-----:|:-----:|:-----:|:-----:|:-----:|:-----:|:-----:|:-----:|:-----:|:-----:|
> > | Ours(S) | 93.73 | 92.41 | 90.08 | 84.40 | 84.07 | 81.37 | 79.92 | 77.30 | 71.93 | 59.34 | 10.00 |
> > | Ours(C)  | 93.53 | 92.67 | 89.99 | 84.47 | 84.20 | 81.41 | 78.42 | 75.71 | 66.90 | 53.00 | 15.36 |

---

> ### Author Response · Authors · 2023-11-20
>
> **Q8:** What about transferability in terms of limited samples used to learn unlearnable examples?
>
> **Ans:** We conducted the transferability study across architectures with limited unlearnable examples, and we show the results in Table F in the supplementary. The test accuracy decreases in a similar trend when we increase the percentage of the unlearnable examples.
>
> **Table F: Test accuracy (\%) of CIFAR-10 on the different models trained by
> the clean data mixed with different percentages of unlearnable examples.**
>
> | Setting                      | Model | 20\%  | 40\%  | 60\%  | 80\%  |
> |:----------------------------:|:-----:|:-----:|:-----:|:-----:|:-----:|
> | Ours(S) | R18   | 93.73 | 92.41 | 90.08 | 84.40 |
> |            Ours(S)                   | R50   | 94.16 | 92.82 | 90.51 | 85.66 |
> |          Ours(S)                     | V19   | 92.11 | 91.14 | 88.77 | 83.48 |
> |           Ours(S)                    | D121  | 89.02 | 87.77 | 84.94 | 81.39 |
> |            Ours(S)                   | ViT   | 75.95 | 74.77 | 74.09 | 70.50 |
> | Ours(C)  | R18   | 93.53 | 92.67 | 89.99 | 84.47 |
> |             Ours(C)                  | R50   | 94.04 | 92.31 | 90.58 | 85.15 |
> |              Ours(C)                 | V19   | 92.13 | 90.23 | 87.99 | 80.60 |
> |               Ours(C)                | D121  | 88.28 | 86.56 | 83.57 | 77.79 |
> |                Ours(C)               | ViT   | 75.44 | 74.58 | 69.34 | 64.80 |
>
> **Q9:** The paper also needs to discuss existing contemporary image-hiding works effective in generating adversarial examples and how they are different from this work. Why they can not be used as compared to the proposed DH?
> ```
> 1. Din SU, Akhtar N, Younis S, Shafait F, Mansoor A, Shafique M. Steganographic universal adversarial perturbations. Pattern Recognition Letters. 2020 Jul 1;135:146-52.
> 2. A. Agarwal, N. Ratha, M. Vatsa and R. Singh, "Crafting Adversarial Perturbations via Transformed Image Component Swapping," in IEEE Transactions on Image Processing, vol. 31, pp. 7338-7349, 2022, doi: 10.1109/TIP.2022.3204206.
> ```
> **Ans:** Thank you for suggesting a comparison with contemporary image-hiding techniques in the context of adversarial examples. Our work indeed shares the image-hiding framework with the studies you mentioned, but our goals and methodologies differ substantially.
>
> The specific aim of our work is to disrupt the learning process of a model. We seek to ensure high training accuracy on the unlearnable examples we generate, while deliberately reducing accuracy on a clean testing set. The adversarial examples crafted in the studies you listed are intended to lead a fully trained model to make incorrect predictions, a different goal than ours.
>
> Additionally, we have expanded our research to different image-hiding networks, notably the ISGAN [D]. Our experiments assess the effectiveness of ISGAN-hidden unlearnable examples. The findings reveal that while unlearnability can be achieved with other deep hiding models like ISGAN, the performance is not as optimal as with our applied Invertible Neural Network (INN). INN demonstrates superior performance in deep hiding [E][F], which is why we chose it as our baseline model to validate our concepts.
>
> We believe these distinctions, along with our comprehensive evaluations, underscore the unique contribution of our work in the field of image hiding and data privacy.
>
> ```
> [D] Zhang, R., Dong, S., & Liu, J. (2019). Invisible steganography via generative adversarial networks. Multimedia tools and applications, 78, 8559-8575.
> [E] Xu Y, Mou C, Hu Y, et al. Robust invertible image steganography. CVPR, 2022: 7875-7884.
> [F] Mou C, Xu Y, Song J, et al. Large-capacity and flexible video steganography via invertible neural network. CVPR, 2023: 22606-22615.
> ```
>
> **Table H: Test accuracy (\%) of model train on unlearnable examples generated by using another deep hiding model (ISGAN).**
> | ISGAN      | Vanilla | Cutout | Cutmix | Mixup | MeanF | MedianF | BDR   | Gray  | GaussN | GaussF | JPEG10 | JPEG50 | AT    | Mean  |
> |------------|---------|--------|--------|-------|-------|---------|-------|-------|--------|--------|--------|--------|-------|-------|
> | Sample-wise| 54.05   | 53.74  | 55.97  | 72.61 | 48.83 | 85.11   | 86.77 | 53.04 | 88.45  | 54.25  | 84.35  | 90.55  | 87.22 | 70.38 |
> | Class-wise| 23.99   | 24.07  | 28.42  | 42.69 | 37.56 | 79.3    | 83.64 | 30.06 | 87.92  | 25.24  | 84.46  | 90.28  | 87.46 | 55.78 |

---

### Meta-Review · Area_Chair_PBHn · 2023-12-05

**Metareview:**

This paper presents an algorithm to construct unlearnable examples that can mislead the deep networks for data protection.

After the rebuttal and AC-reviewer discussion stage, the final scores of this paper are 5/5/5/6. One reviewer changed his/her score from 6 to 5 after the discussion. The only positive reviewer (rating 6) did not show up in the discussion, while the other three reviewers arrived at a consensus of rejection. The AC found no reason to overturn the reviewers' recommendation.

**Justification For Why Not Higher Score:**

The reviewers arrived at a consensus of rejection.

**Justification For Why Not Lower Score:**

N/A

---

### Decision · Program_Chairs · 2024-01-16

Reject